# Research

palaeontology, evolution

evolution, socio-sexual selection, dinosauria, geometric morphometrics

**Author for correspondence:**
A. Knapp
e-mail: a.knapp@nhm.ac.uk

# Three-dimensional geometric morphometric analysis of the skull of *Protoceratops andrewsi* supports a socio-sexual signalling role for the ceratopsian frill

A. Knapp[1,2], R. J. Knell[2] and D. W. E. Hone[2]

[1]Department of Earth Sciences, The Natural History Museum, London, UK
[2]Department of Organismal Biology, Queen Mary University of London, London, UK

AK, 0000-0003-4822-5623; RJK, 0000-0002-3446-8715

Socio-sexual selection is predicted to be an important driver of evolution, influencing speciation, extinction and adaptation. The fossil record provides a means of testing these predictions, but detecting its signature from morphological data alone is difficult. There are, nonetheless, some specific patterns of growth and variation which are expected of traits under socio-sexual selection. The distinctive parietal-squamosal frill of ceratopsian dinosaurs has previously been suggested as a socio-sexual display trait, but evidence for this has been limited. Here, we perform a whole-skull shape analysis of an unprecedentedly large sample of specimens of *Protoceratops andrewsi* using a high-density landmark-based geometric morphometric approach to test four predictions regarding a potential socio-sexual signalling role for the frill. Three predictions—low integration with the rest of the skull, significantly higher rate of change in size and shape during ontogeny, and higher morphological variance than other skull regions—are supported. One prediction, sexual dimorphism in shape, is not supported, suggesting that sexual differences in *P. andrewsi* are likely to be small. Together, these findings are consistent with mutual mate choice or selection for signalling quality in more general social interactions, and support the hypothesis that the frill functioned as a socio-sexual signal in ceratopsian dinosaurs.

## 1. Introduction

Sexual selection, arising from competition for fertilization opportunities, is the evolutionary force responsible for many of the more extraordinary features that we see in extant animal taxa: the elaborate feathers and courtship displays of birds of paradise, the diversity of horns carried by bovids, the enlarged mandibles of stag beetles and many more [1]. Sexual selection is predicted to be a powerful driver of many important evolutionary processes at levels above individual morphology, including speciation [2–4] and adaptation to new environments [5,6].

Because sexual selection concerns intraspecific competition for resources, it can be considered a subset of a broader phenomenon, social selection [7], which may also drive the evolution of weapons, ornaments and behaviours in both sexes to establish dominance hierarchies within populations [8,9]. The term 'socio-sexual selection' thus encompasses a range of closely linked evolutionary drivers. Given this potential role in macroevolutionary processes, identifying the signature of socio-sexual selection in the fossil record would be a crucial step in understanding how it can shape evolution [10].

The large, ornamented skulls of ceratopsian dinosaurs have no obvious analogue in living taxa, and the function of the diverse horns and frills borne on them is still a cause for debate [11–14]. Socio-sexual selection is a strong candidate for the driver of skull evolution in ceratopsians [12,13] but has been rejected by some researchers due to the apparent lack of sexual dimorphism in any ceratopsian taxa [11]. It is possible, however, that this apparent lack of dimorphism is a consequence of insufficient sampling, a perennial problem when studying extinct animals [15], or the result of socio-sexual selection for mutual ornamentation [8,16–18].

Given the lack of any one truly diagnostic test for sexual or socio-sexual selection in the morphology of extinct taxa, it is important to consider several lines of evidence when assessing the roles of putative signal traits [10,19]. Previous studies of possible socio-sexual traits in extinct taxa have been limited in this regard, partly because of their use of fairly simple criteria based on, for example, linear measurements or qualitative descriptive approaches. Here, we employ three-dimensional (3D) geometric morphometric (GM) techniques, enabling a more comprehensive analysis of shape [20]. This approach allows us to analyse a possible signal trait in enough detail to allow a number of predictions regarding the potential signal of socio-sexual selection to be tested. These are as follows.

## (a) Low morphological integration with other regions of the skull

Phenotypic modularity, the formation of internally integrated sets of traits which are only loosely integrated with other sets of traits, allows the developmental and evolutionary independence of distinct regions upon which selection may act [21–23]. If a trait functioned as a socio-sexual signal, we predict that it should form a distinct phenotypic module, allowing selection to drive the evolution of its form without impeding the development and function of other regions of the organism. Phenotypic modules can be identified by comparing patterns of morphological covariance across traits [21,22] and can serve as the basis for further comparisons of growth and variation [24,25]. Modules linked to sexually selected cranial ornamentation in extant squamates have been shown to exhibit heightened morphological disparity and evolutionary rates [26]. Any set of traits that combines to perform an adaptive function may form phenotypic modules and may undergo strong selection; therefore, analysis of shape variation between different modules can help to provide further information about a module's function [21,24].

## (b) Significant change during ontogeny and positive allometry

Positive allometry, an increased relative change in size and shape, is typical of many socio-sexual traits during growth (ontogenetic allometry), and among individuals of the same species and developmental stage (static allometry) [19,27]. Ontogenetic allometry occurs because as animals become sexually mature the importance of socio-sexual traits changes, meaning that the optimal form will vary accordingly with the animal's growth [27,28], and probably evolves through heterochrony [29]. Positive static allometry is expected to occur in a socio-sexual trait when the relative size of the trait in question is an 'honest' signal of condition or resource-holding potential,

resulting in small differences in body size being reflected in proportionally larger differences in trait size [19]. Positive allometry is known from socio-sexual traits in a wide range of extant taxa, including male jaw shape of the neotropical electric fish genera *Apteronotus* and *Compsaraia* [30], eyestalk span in the stalk-eyed fly genus *Teleopsis* [10] and antler size in the white-tailed deer *Odocoileus virginianus* [19]. Both ontogenetic and static allometry are related and leave a similar signal which may be readily detected in a large sample. Positive allometry alone is not necessarily diagnostic of a function in sexual or socio-sexual signalling since it has also been reported from a small number of non-signal traits, one example being the femur length of the water strider *Gerris buenoi* [27]. The degree of positive allometry shown by socio-sexual traits is usually, however, substantially higher than in other naturally selected traits, even those which themselves show positive allometry [19].

## (c) High variance

The 'honest' nature of socio-sexual signals is often maintained by these traits showing strong 'condition-dependence', whereby they respond more strongly to the overall health and well-being of the bearer than do other traits [19,31]. Without specific functional constraints, we would expect socio-sexual traits to show markedly higher intraspecific variance than other traits. This is found in socio-sexual traits in a wide range of extant taxa from the dung beetle genus *Onthophagus* to bovids such as *Connochaetes* [19]. When dealing with fossil datasets, it is also important to consider the effects of taphonomic deformation, which may increase morphological variance [32].

## (d) Sexual dimorphism

Sexually selected traits in extant taxa often show marked sexual dimorphism because of the different selective pressures acting on the sexes; hence, strong sexual dimorphism is often an important indicator of sexual selection and is well known across a wide range of extant taxa [1]. As with our other criteria, sexual dimorphism or its absence is not itself diagnostic of sexual selection; however, traits can be sexually dimorphic for reasons other than sexual selection [10], and a lack of obvious dimorphism does not rule out a sexual function for a trait [17]. Furthermore, if a trait functions as a signal in socio-sexual contexts [7,8], or if mutual sexual selection operates in the population [16], then sexual dimorphism may be negligible. The detection of sexual dimorphism in fossil taxa, where sex cannot be determined independently, is especially difficult and requires large sample sizes if the effect is small [15].

The ceratopsian dinosaur *Protoceratops andrewsi* possessed a large frill projecting caudally from the skull, formed from enlargement of the parietal and squamosal bones (electronic supplementary material, figure S1). This frill has previously been suggested as a potential sexual or socio-sexual signal, and positive allometry [13], modularity [29] and sexual dimorphism [33,34] have all been assessed using different measurements. *Protoceratops andrewsi*, being one of the few dinosaur examples with sufficient well-preserved specimens available to allow an in-depth analysis of morphology and variation, is an ideal candidate for examining intraspecific variation in non-avian dinosaurs, but a clear picture of the function of the frill has yet to emerge. Incorporating additional

specimens allows us to greatly expand the size of our dataset over previous studies, and using a high-density GM approach allows us to assess the morphology of the entire skull of *P. andrewsi* with a focus on 3D shape rather than linear measurements or two-dimensional (2D) shape as in these previous studies [35]. This approach enables us to address the four hypotheses outlined above with a single dataset.

## 2. Methods

### (a) Data collection

Three-dimensional digital mesh models of 65 *P. andrewsi* skulls were created using photogrammetry software [36], with between 50 and 200 photographs each depending on specimen accessibility. Meshes were decimated to 200 000 polygons before landmarking. Six specimens with small amounts of asymmetric taphonomic deformation, defined as a deviation of the sagittal plane from vertical relative to the rest of the skull of less than 10°, were retrodeformed in Landmark v. 3.0 [37] using established protocols [38] (electronic supplementary material, table S1). Specimens that were severely taphonomically deformed were omitted from the dataset. Of the remainder, 30 skulls were complete enough to be used for whole-skull analyses (electronic supplementary material, table S1).

We used a landmark-based approach to define anatomical points, sutures and surface morphology [39]. Landmarks were placed on the right-side upper surface of the skull with Landmark v. 3.0 [37]. Specimens that were damaged or incomplete on the right side were mirrored and landmarks were placed on the reflected left side. A total of 821 landmarks and semilandmarks were placed on each skull (electronic supplementary material, table S2 and figure S2), including 299 surface semilandmarks placed using a semi-automated approach with the *Morpho* package v. 2.8 for R [39–41]. Semilandmarks were then slid to minimize bending energy [20]. Variation in the width of specimens may shift the Procrustes alignment away from the midline of a bilaterally symmetrical structure such as a skull, affecting the accuracy of analyses performed on shape data aligned in this way [42]. All landmarks were thus reflected across the sagittal plane of the skull before a generalized Procrustes alignment was performed with the *geomorph* package for R [43]. After alignment, reflected landmarks were removed and all subsequent analyses were performed on the Procrustes-aligned right-side landmarks.

All fossil specimens are potentially subject to taphonomic deformation [32]. The dilation component of taphonomic deformation describes symmetrical plastic deformation, resulting in dorsoventral or lateral compression of specimens. We examined projected shapes created from principal component analysis outputs of Procrustes-aligned data to identify and estimate this component of shape variation [44].

### (b) Statistical analysis

Allometry was explored by calculating standardized shape scores from the regression of shape on whole-skull centroid size [20,43]. The 'common allometric component' (CAC) is the component of shape variation most closely aligned with size and allows the removal of residual shape variation [45].

Modularity hypotheses were explored with a maximum-likelihood (ML) method implemented with the R package *EMMLi*, which compares a number of different module hypotheses [22]. A total of 20 module hypotheses were tested (electronic supplementary material, table S4) from a two-module to an eight-module hypothesis. Analyses were performed with shape data both corrected and uncorrected for allometry [46]. High-density landmark coverage tends to increase support for higher

numbers of modules in ML analyses [25], and to compensate for this we merged modules where the between-region correlation value was within 0.1 of the lowest within-region correlation value [25]. In addition to the ML approach, we further assessed modularity by calculating the covariance ratio (CR) for both the original and allometry-corrected datasets [43].

Ontogenetic variation was assessed in two ways:

(1) Individual module centroid sizes were regressed against whole-skull centroid size for each specimen to give a measure of relative size change with ontogeny for each module. Regression slopes were compared between different modules using an ANCOVA [19]. This method is similar to the method employed in previous studies [13,19] but compares all skull modules.
(2) CAC values of each globally aligned module were regressed against whole-skull centroid size [30]. This method gives a relative measure of degree of shape change for all modules; no change in shape results in a slope of 0. Module slopes were compared using an ANCOVA.

Procrustes variances were calculated for each individual landmark and for all modules. The variance of each module was divided by the number of landmarks for that module to provide comparable results [25].

To assess sexual dimorphism, we assumed bimodal distribution of shape data and tested for non-unimodal distribution with Hartigan's dip test, which measures multimodality by comparing the empirical sample distribution function with a unimodal distribution function that minimizes the maximum difference [47]. Dip tests were performed on the first eight residual shape components of allometry-corrected datasets for whole-skull shape data, plus separate tests for each module both globally and separately aligned. An additional 14 partial specimens were included in the analyses of individual modules where the relevant modules were present in these specimens (electronic supplementary material, table S1). Dip tests were repeated with the smallest, 'juvenile' specimens removed. These specimens were identified from the allometry plot (electronic supplementary material, figure S7) as grouping into two distinct clusters, separated from the cluster of larger 'adult' specimens via shape mean.

Additional analyses were run with modifications to the dataset to address possible biases due to specimen location and deformation (see electronic supplementary material for discussion on specimens and additional analyses).

## 3. Results

### (a) Modularity

The *EMMLi* analysis gave strongest support for the eight-module hypothesis. Merging modules with high between-module correlations [25] resulted in five separate modules: frill, postorbital, nasal–premaxilla–rostral (hereafter 'snout'), maxilla and jugal (electronic supplementary material, figure S6). CR analysis supported a significantly modular skull structure (CR = 0.86 for raw data; CR = 0.67 for allometry-corrected data, $p < 0.01$; electronic supplementary material, tables S5 and S6).

### (b) Whole-skull shape variation

The first three PCs account for 79.8% of cumulative shape variation, and 95% is explained by the first 11 PCs (electronic supplementary material, figure S5). There is a significant correlation between skull size and shape in *P. andrewsi* ($p \leq 0.01$,

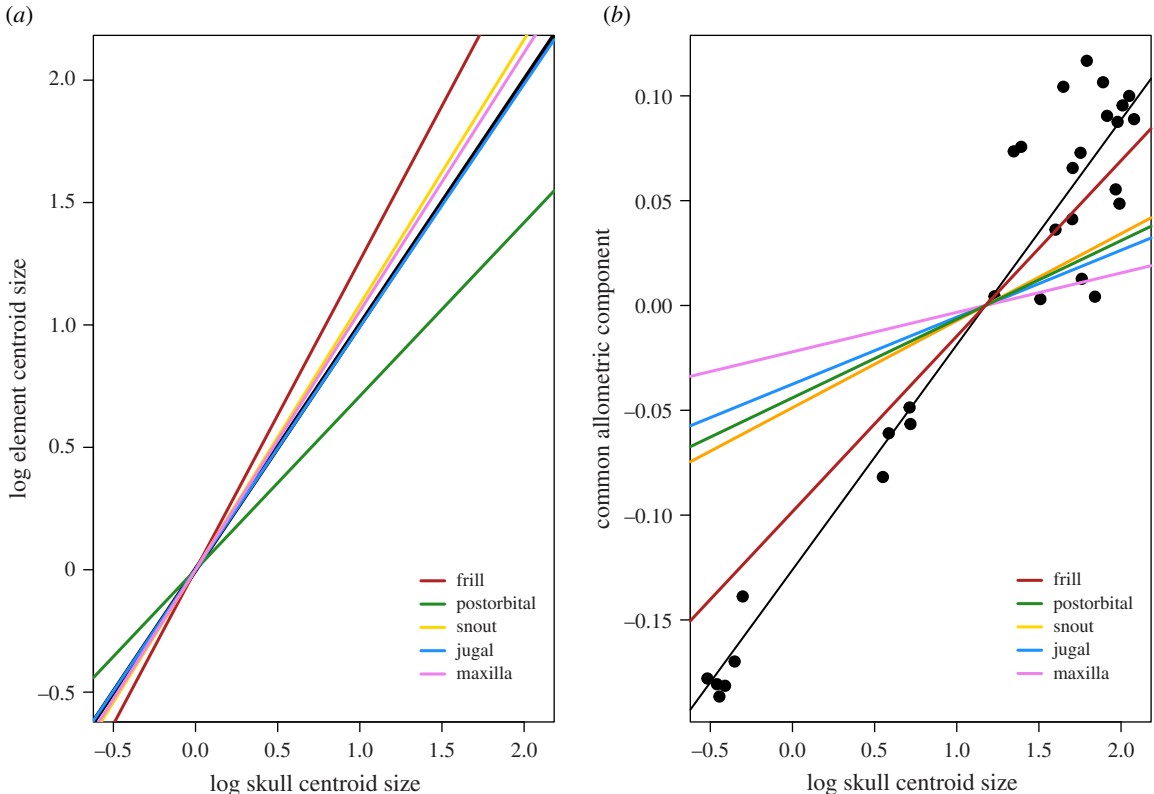

**Figure 1.** Log centroid size values of individual modules regressed against log skull centroid size (*a*) and CAC regressed against log skull centroid size (*b*). The black line in (*a*) is a slope of 1, representing isometry. Intercepts for all modules in (*a*) are set to zero to standardize comparisons. Black points in (*b*) represent whole-skull shape data, and the black line is the regression for these data. Individual module regression lines are coloured according to the module (individual specimen points not shown for separate modules). (Online version in colour.)

$R^2 = 0.52$; electronic supplementary material, figure S9), indicating that around 52% of total shape variation can be accounted for by allometry. PC4 (4.9% of total shape variation) shows obvious signs of taphonomic deformation, i.e. flattening of the skull and deformation of the orbit [32] (electronic supplementary material, figure S6). Although arguably not as clear in PCs 2 and 3, it is likely that effects of taphonomic deformation are distributed more subtly among these and subsequent PC axes.

### (c) Changes with skull size: relative module size

As suggested by previous studies [13,19], the frill shows the highest rate of size change during ontogeny (figure 1*a*), with an allometric slope that is significantly greater than that of the next-highest module, the snout ($p \leq 0.0001$; electronic supplementary material, table S7). By contrast, the difference between the rate of growth of the snout and the maxilla was not significant ($p = 0.32$). The postorbital, incorporating a large portion of the orbit margin, shows the lowest change in module size with respect to skull size.

### (d) Changes with skull size: module shape

The frill also displays the greatest change in shape with size of all modules (figure 1*b*), significantly steeper than that of the second-highest module, the snout ($p \leq 0.0001$). The difference in relative rates of shape change between the snout and the postorbital was not significant ($p = 0.49$). Projected shape changes of individual modules between minimum and maximum skull sizes are shown in figure 2, ranked in descending order of relative shape change from greatest (frill) to least (maxilla).

### (e) Morphological variation

When corrected for allometry, individual module disparity was found to be highest in the frill ($1.27 \times 10^{-5}$) and jugal ($1.05 \times 10^{-5}$), and lowest in the postorbital ($4.12 \times 10^{-6}$; electronic supplementary material, figure S8). No significant relationship was found between Procrustes variance (morphological disparity) and either growth rate (slope of module centroid size regressed against whole-skull centroid size; electronic supplementary material, figure S8A) or within-module correlation values (derived from *EMMLi* analysis; electronic supplementary material, figure S8B). Relative per-landmark variation across the entire skull is displayed in electronic supplementary material, figure S9, for both raw and allometry-corrected datasets.

### (f) Sexual dimorphism

No dip test was below the 5% significance level for any shape component for either whole-skull data, or for any individual module (electronic supplementary material, table S8), for both the dataset containing all specimens and for the dataset with 'juveniles' removed. Only component 6 of the whole-skull shape data for all specimens returned a *p*-value approaching 0.05 ($p = 0.09$; $R^2 = 0.04$). Because this shape component contributes only approximately 2.5% of total skull shape variation, it is unlikely to represent true population-level dimorphism.

## 4. Discussion

Of the four predictions outlined in the Introduction, three were supported by this study. Evidence was found in *P. andrewsi* for

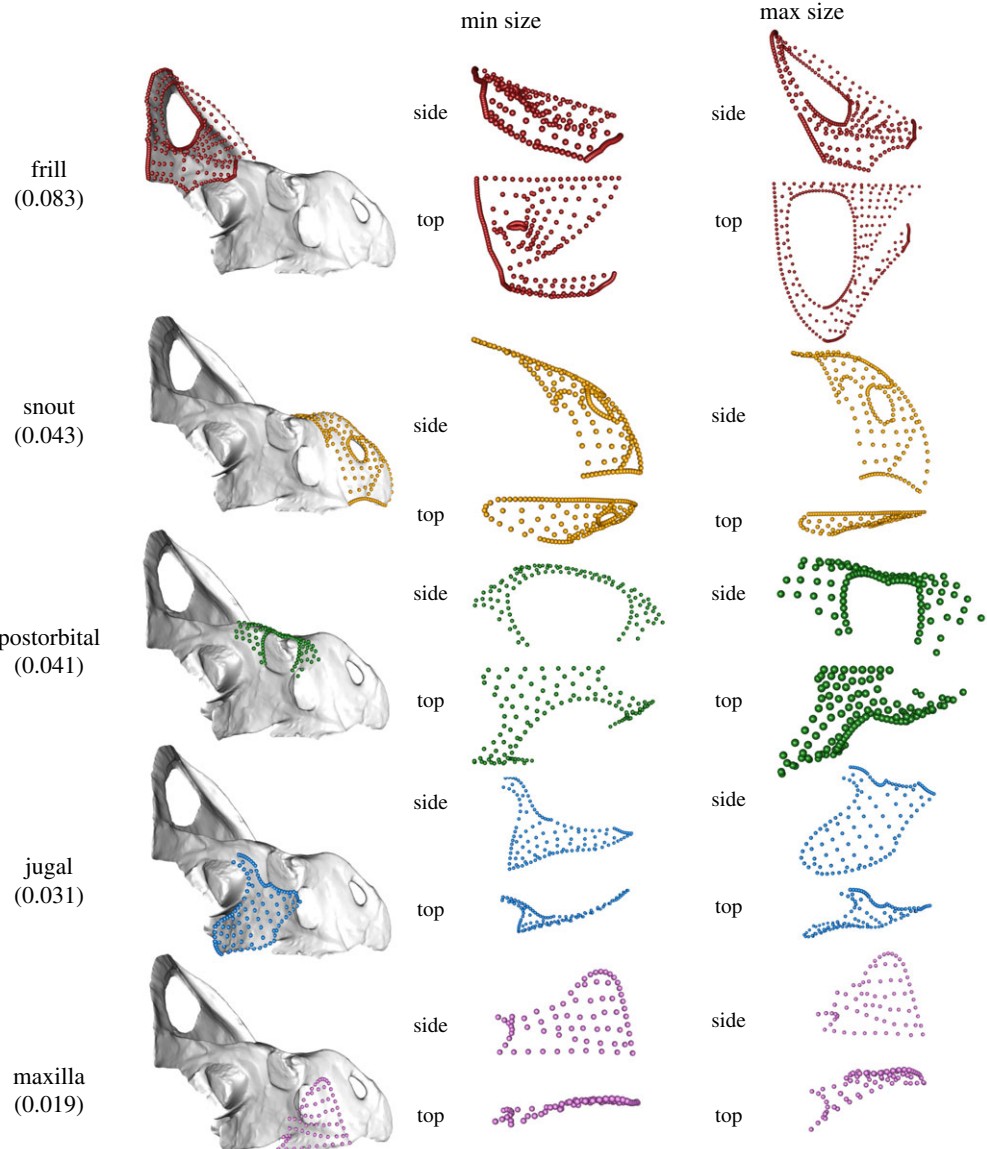

min size        max size

**Figure 2.** Projected shapes at minimum (Min) size and maximum (Max) size of the five modules identified in *EMMLi*. Landmarks for each module are superimposed on the mesh model at left (AMNH 6439). Shapes of individual modules at maximum and minimum projected specimen sizes are shown at right (not to scale). Modules are ranked from highest to lowest relative shape change. Slope values, derived from CAC regressed against centroid size (figure 1*b*), are shown in parentheses below the module name (whole-skull slope = 0.108). All individual modules are shown in right-lateral (side) and dorsal (top) views for both sizes. (Online version in colour.)

modularity in the skull (including the frill forming a distinct module), significantly higher rates of change in size and shape of the frill during ontogeny and higher morphological disparity in the frill independent of size. One prediction, that the frill would show sexual dimorphism, was not supported by this study. On this basis, we conclude that our results provide further evidence that the exaggerated parietal-squamosal frills of ceratopsian dinosaurs fulfilled a socio-sexual signalling role. Individuals of *P. andrewsi* probably used the frill as a visual signal to assess conspecifics as potential rivals in dominance hierarchies and/or as possible mates.

## (a) Prediction 1: modularity

We found strong support for a five-module model of the cranium of *P. andrewsi*. Although the treatment of the frill as a distinct module [29,34] is supported by our analysis, it is clear from these results that cranial modularity of *P. andrewsi* is more complex than a simple division of frill and remainder of the skull. Modularity suggests that regions of the skull

independently vary in their rate of growth, ontogenetic shape change and morphological variance. The identification of discrete phenotypic modules in *Protoceratops* reveals a developmental framework by which skull elements can differentiate and become co-opted for different purposes [21,23]. Traits that have a signalling function typically show more variety of form and accelerated rates of development compared with non-signalling traits [19], and as distinct modules they are able to semi-independently develop and respond to selection [21].

The use of high-density 3D data to examine evolutionary modularity has been employed in a number of extant and extinct clades [23,25,26], but analyses of ontogenetic modularity are less common [46]. The skull of *P. andrewsi* appears to show higher skull integration than the pattern seen across archosaurs in general [23]. The bones of the postorbital module (lachrymal, frontal, prefrontal and postorbital) [48] fuse in many specimens of *P. andrewsi*, making them difficult to landmark individually. Our *EMMLi* analysis revealed this region to be relatively highly integrated, justifying the treatment of this

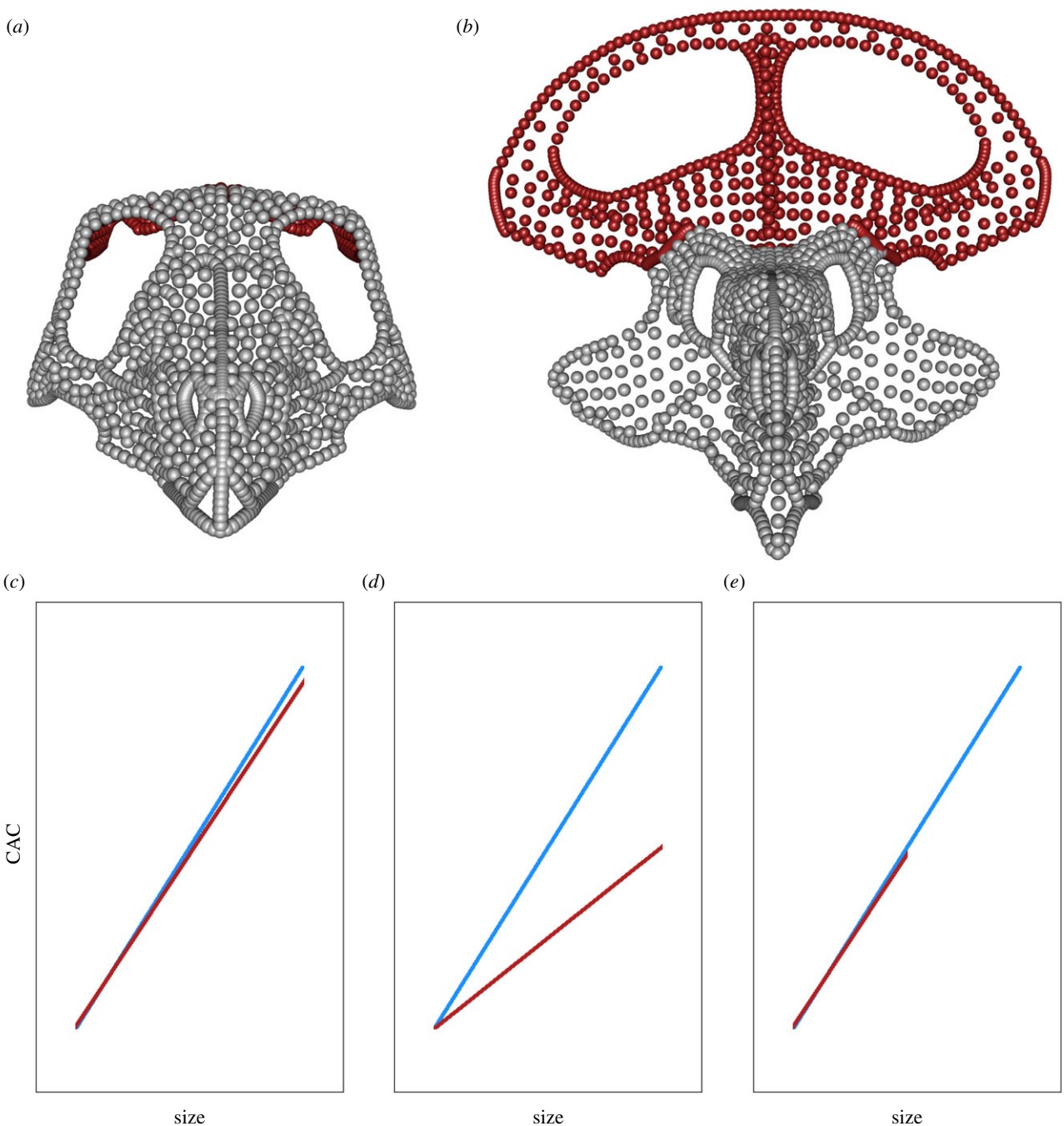

**Figure 3.** Anterior view of projected skull shapes at minimum (*a*) and maximum (*b*) size (not to scale), with frill coloured red to illustrate the difference in both size and shape from these two extremes. (*c*–*e*) show hypothetical examples of growth trajectories under different forms of sexual dimorphism (CAC against size; cf. figure 1*b*) of two sexes. (*c*) Sexually monomorphic; minimal difference in size and shape between sexes. (*d*) Sexual shape dimorphism; sexes grow to similar size but different shape. (*e*) Sexual size dimorphism; sexes follow same growth trajectory but one sex grows to larger size and shows greater shape change as a result. (Online version in colour.)

region as a single module. Integration is also apparent in the frill and snout regions, which might be expected given the unique skull morphology of ceratopsians [48]. Accordingly, we may expect similar patterns of integration to be seen across other ceratopsian taxa if homologous traits performed similar functions across the clade.

## (b) Predictions 2 and 3: positive allometry and morphological disparity

In common with previous studies, size was found to be strongly correlated with whole-skull shape in *P. andrewsi* [13,32,33], accounting for around 52% of total shape variation

in this dataset and reflecting drastic changes in skull shape during ontogeny [28]. Although all modules showed some degree of change, the parietal-squamosal frill consistently displayed the highest rates of size and shape change through ontogeny, as well as the highest morphological variance independent of size, expanding significantly during ontogeny to form a large, conspicuous structure (figure 3*a,b*). The accelerated growth rate of the frill when compared with the remainder of the skull adds further support to the hypothesis that this structure evolved as a socio-sexual signal, and probably evolved via heterochrony [29]. There are two important differences between our allometry analyses and those in previous studies [13,19,33]. First, we compared the allometric

slopes of five separate, statistically supported modules rather than only the rate of change of frill size with skull size. Second, we analysed 3D shape allometry for the first time in a dinosaurian taxon.

Regarding the other skull modules, the dorsal surface of the nasals shows slight bulging in the largest specimens but not to the extent of forming a distinct, fused nasal horn as in related taxa [49]. It is possible that this thickened region functioned in intraspecific combat, as some authors have suggested [33], but testing this hypothesis would require structural analysis of the nasals to determine if they have the strength necessary to function as weapons [50], or evidence of keratinous covering consistent with horns in other taxa [48]. The jugal has also been suggested as a secondary sexual trait in *Protoceratops* [13]; prominent jugal 'horns' are a characteristic feature of numerous ceratopsian taxa [48]. The data presented here are ambiguous on this issue: although the jugal shows the second-highest allometry-adjusted morphological disparity, it does not fit the patterns of either heightened growth rate or shape change. Many extant taxa bear multiple sexually selected signals, but their relative expression may be influenced by different factors [51]. Quantifying evolutionary rates across a range of ceratopsian taxa will enable patterns of evolution of different cranial modules to be explored and may provide a clearer picture of the role of jugals in ceratopsians [24].

Of the three supported hypotheses, morphological disparity is most difficult to interpret. As noted by Dodson [33], variance is highest in the frill, a prediction of socio-sexually selected traits [19]. No pair of any traits would be expected to have identical levels of variance, but we would expect to see a predicted socio-sexual trait to display notably higher variance than any other non-sexually selected trait, as we do here. Although we corrected minor asymmetrical deformation and omitted more severely deformed specimens, taphonomic deformation is likely to contribute to increased disparity across the skull [32,52]. Taphonomic deformation may also affect modularity estimates by increasing correlation values across the skull, because compaction forces from overlying sediment act in the same direction across it. It is therefore possible that these values are overestimated [35]. The large (approx. 52%) contribution of size-correlated shape data on the dataset, combined with the lower between-module correlation values once allometry is corrected for suggests that, while undoubtedly present, taphonomic deformation does not overwhelm biological signal, even when the effects of size are removed [32] (electronic supplementary material, tables S5 and S6).

## (c) Prediction 4: sexual dimorphism

No support was found for sexual shape dimorphism in the skull of *P. andrewsi* in this study, either in individual modules or in whole-skull shape data. In this study, we have focused on analysing 3D shape data, but previous attempts to determine sexual dimorphism in *P. andrewsi* using linear measurements, descriptive observations and 2D GMs have produced mixed results [33,34,53]. In contrast to previous studies, we did not presuppose what or where sexual differences might be, but tested for non-unimodality in shape variation across the entire skull. Without pronounced shape differences creating distinct allometric trajectories between sexes (figure 3*d*), sexual dimorphism can be difficult to

identify without prior knowledge of sex, even with a very large sample size [15], because the relevant statistical tests are often subject to type II error [53]. The strong correlation between shape and size suggests that the skull shape of *Protoceratops* is either not strongly sexually dimorphic in shape and any differences are masked by other sources of shape variation (figure 3*c*), or that males and females share a similar growth trajectory but are dimorphic in size (figure 3*e*). It is possible that a larger dataset would resolve this issue [53].

Despite its importance in Darwin's original work on sexual selection [1], the lack of detectable dimorphism in this dataset is not fatal to the hypothesis that the frill of *P. andrewsi* acted as a socio-sexual signal for two main reasons. First, the magnitude of sexual dimorphism can vary greatly between species and can, in some cases, be negligible [17]. Selection for sexual dimorphism may be purely sexual [16], a result of more complicated socio-sexual selection [8], or may be due to ecological niche differentiation between sexes [54,55]. Second, sexual dimorphism in extant organisms is often most obvious in either body size, which we did not analyse here, or in soft tissues, such as feathers and coloration [51], which are rarely fossilized and may not be detectable in underlying skeletal morphology [56]. Given the widespread prevalence of some form of sexual dimorphism in extant taxa, including archosaurs, it is likely that *Protoceratops* was indeed sexually dimorphic to some degree [53]. Our findings show that any dimorphism present does not, however, appear to be in the form of conspicuous shape differences in the skull and, in particular, the frill, and the fact that all known individuals of *P. andrewsi* possessed a frill suggests a role for this distinctive trait in both sexes. Other functional explanations for the ceratopsian frill, such as a defensive, thermoregulatory or species recognition structure, have little support [11,14,33]. Without the structural and functional constraints of mechanical traits, socio-sexually selected traits are free to vary much more readily within a population but will be opposed by natural selection if they are not selected for [17], so it is unlikely that a seemingly costly trait would be maintained across ceratopsians without an explicit function.

Using a high-dimensional GM approach, we have shown that the frill of *P. andrewsi* shows several characteristics consistent with a socio-sexual trait. Socio-sexual selection is predicted to be a powerful driver of evolution, potentially resulting in mutual ornamentation and runaway selection under certain circumstances [8,16–18]. Identifying socio-sexual signal traits in an extinct taxon has implications for not only better understanding the palaeobiology of that taxon [57], but also the evolutionary implications of socio-sexual selection [10]. If the frill of *Protoceratops* functioned as a socio-sexual signal of dominance and/or individual genetic quality, as we suggest, it implies that intraspecific social interaction of some kind was important in this taxon [7,8]. Confirming sexual selection would ultimately rely on establishing a connection between trait expression and reproductive success [58], an impossibility in any fossil taxon. Nonetheless, the distinction between sexual and social selection may not be important, because both result from intraspecific competition for resources and are thus intimately connected [7,17]. Socio-sexual traits are expected to show rapid rates of evolution [3,9,10], and extending similar morphometric methods to other ceratopsian taxa would allow us to test this hypothesis with analyses of modularity and putative socio-sexual trait

evolution [24], providing further evidence of socio-sexual selection within an evolutionary framework.

**Competing interests.** We declare we have no competing interests.

**Funding.** This work was supported by the Natural Environment Research Council (NERC) doctoral training grant no. (NE/L002485/1) and by a Jurassic Foundation grant.

**Acknowledgements.** We thank Ryan Felice (UCL/NHMUK), Sandra Álvarez Carretero (QMUL) and Anjali Goswami (NHMUK/UCL) for helpful discussions and advice with data analysis. Scott Hartman provided the skeletal drawing in electronic supplementary material, figure S1. For access to and help with specimens, we wish to thank Mark Norell and Carl Mehling (AMNH), Khishigjav Tsogbaatar, Purevdorj Khataanbaatar and Nyamkhishig Tsogjargal (MAS), Sandra Chapman, Susie Maidment and Tom Raven (NHMUK), Jolanta Koblylinska, Justyna Słowiak and Łukasz Czepinski (ZPAL), and Matt Lamanna and Amy Henrici (CMNH). Jordan Mallon, Caleb Brown and two anonymous referees are thanked for comments on an earlier version of this paper.

**List of institute abbreviations.** AMNH: American Museum of Natural History, New York, USA. CMNH: Carnegie Museum of Natural History, Pittsburgh, USA. MAS: Mongolian Academy of Sciences, Ulaanbaatar, Mongolia. NHMUK: The Natural History Museum, London. QMBC/QMUL: School of Biological and Chemical Sciences, Queen Mary University of London, UK. UCL: University College London, UK. ZPAL: Institute of Paleobiology, Polish Academy of Sciences, Warsaw, Poland.

**Data accessibility.** Morphological landmark data for specimens is included in the electronic supplementary CSV files. All analyses were performed in R using existing packages, referenced in the text.

**Authors' contributions.** A.K., R.J.K. and D.W.E.H. conceived the study. A.K. gathered and processed scan data, and ran analyses. A.K., R.J.K. and D.W.E.H. wrote the manuscript.

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
