## [Reviewer comments · Proceedings of the Royal Society B: Biological Sciences]

Review History

RSPB-2020-1129.R0 (Original submission)

Review form: Reviewer 1

Recommendation

Major revision is needed (please make suggestions in comments)

Scientific importance: Is the manuscript an original and important contribution to its field?

Good

General interest: Is the paper of sufficient general interest?

Good

Quality of the paper: Is the overall quality of the paper suitable?

Good

Is the length of the paper justified?

Yes

Should the paper be seen by a specialist statistical reviewer?

No

Do you have any concerns about statistical analyses in this paper? If so, please specify them explicitly in your report.

No

It is a condition of publication that authors make their supporting data, code and materials available - either as supplementary material or hosted in an external repository. Please rate, if applicable, the supporting data on the following criteria.

Is it accessible?

Yes

Is it clear?

Yes

Is it adequate?

Yes

Do you have any ethical concerns with this paper?

No

Comments to the Author

Knapp et al., Review Three-dimensional geometric morphometric analysis of the skull of *Protoceratops andrewsi* supports a socio-sexual signalling role for the ceratopsian frill
The authors examined the skull of *Protoceratops andrewsi* using three-dimensional geometric morphometrics in order to assess whether the frill could be considered to have a role in socio-sexual signaling. To this end, they employed modularity analyses (EMMLi), tested for ontogenetic and static allometry, and examined morphological disparity of modules uncovered in their EMMLi analysis. They found that the *Protoceratops* skull is highly modular (8 modules) and found evidence that the frill evolved for socio-sexual signaling, but did not find evidence of sexual dimorphism. This is a strong paper and is well written. I particularly liked the structure of the paper (sets of predictions in the introduction mirrored in the discussion). The figures are also quite effective. I have outlined major comments and line-by-line comments below.

Comments:

I would like to see more of a discussion of taphonomic distortion. I have seen numerous *Protoceratops* skulls and none of them are completely without taphonomic distortion (many are quite distorted). The authors should include a discussion of the distortion in the discussion (likely between the penultimate and ultimate paragraphs). I understand that you did use the retrodeform function in landmark and then considered only 5% of total variation to be taphonomic with PC4 as your 'taphonomy PC'. However, there is not a single taphonomic distortion PC. All PCs will have some degree of taphonomic input. Hedrick et al. (2019) recently found that this can be as high as 40% even in relatively undeformed fossils. Tschopp et al. (2013) similarly found that distortion was extremely difficult to correct and the retrodeformation function in landmark makes the data more symmetric, but does not remove taphonomy from the equation. Hedrick and Dodson (2013) looked at 'relatively undeformed' psittacosaur skulls, but still found taphonomy to be a major driver in shape space. Not to mention that there is an unknowable amount of symmetric plastic deformation that may be affecting your skulls. This certainly does not invalidate your analysis, but these issues must be discussed. Please expand on this. In addition to those above and Angielczyk and Sheets (2007), you can also look at Baert et al. (2014) and Arbour and Currie (2012).

More methodological details are needed. Piecing the methods and results together, I do not think that there are any issues. However, the methods are lacking in information. Here are a few examples: How was the photogrammetry performed? How were models decimated? How were semi-landmarks slid along surfaces (bending energy or Procrustes distance)? How was the

dilation component of taphonomic deformation identified and quantified using PCA?

I would also like to see more of a discussion of sexual size dimorphism. Though osteological sexual shape dimorphism is sometimes found in extant taxa, sexual size dimorphism is much more common. Please expand this section, which may require additional analyses.

Please add an EMLi table with all modularity hypotheses into the main text.

It would be great to expand on the discussion of modularity (lines 212-221) with extant examples. This is the first study of integration and modularity using GM in dinosaurs to my knowledge, but there is a rich extant literature. I think that expanding this part of the paper will make the paper more citable and expand its breadth to workers outside of paleobiology.

Line by line:

Line 19: What do you mean by 'high-dimensional'? 3D? Also line 102 and 284.

Line 20: Maybe 'Three predictions'? Three by itself reads strangely.

Line 25: By broader social contexts, do you mean species recognition?

Line 44: delete 'both'

Line 64: Modularity needs to be more clearly defined. It is not just that they are integrated sets of traits. Modules are highly integrated within themselves, but loosely integrated with adjacent modules. Please also add more citations to this. Although Klingenberg (2008) is one of the first papers to use modularity with GM, it is not the only important citation. Add Adams and Collyer (2016) and additional citations.

Line 69: 'as the basis for further'

Line 118: Are the results substantially different when you do not reflect landmarks? I have not seen this done before. Many datasets in the literature have GPA done on only one side.

Line 127: Modules are hypotheses and they have not yet been identified. Perhaps say hypothesized modules here?

Line 236: True; also, the nasal horn was likely covered in a keratinous sheath of unknown dimensions.

Figure 1: I find it a little confusing to have the dashed black line be isometry in 1A, but the trend line in 1B. Can you make the line in 1B a solid black line?

Adams, D.C. and M.L. Collyer. 2016. On the comparison of the strength of morphological integration across morphometric datasets. *Evolution*. 70:2623-2631.

Hedrick, B.P. and Dodson, P., 2013. Lujiatun psittacosaurids: understanding individual and taphonomic variation using 3D geometric morphometrics. *PLoS One*, 8(8).

Tschopp, E.D., Russo, J. and Dzemplski, G., 2013. Retrodeformation as a test for the validity of phylogenetic characters: an example from diplodocid sauropod vertebrae.

Hedrick, B.P., Schachner, E.R., Rivera, G., Dodson, P. and Pierce, S.E., 2019. The effects of skeletal asymmetry on interpreting biologic variation and taphonomy in the fossil record. *BRANDON P. HEDRICK ET AL. BIOLOGIC VARIATION IN THE FOSSIL RECORD. Paleobiology*, 45(1),

pp.154-166.

Arbour, V. M., and P. J. Currie. 2012. Analyzing taphonomic deformation of ankylosaur skulls using retrodeformation and finite element analysis. PLoS ONE 7:e39323.

Baert, M., M. E. Burns, and P. J. Currie. 2014. Quantitative diagenetic analyses of *Edmontosaurus regalis* (Dinosauria: Hadrosauridae) postcranial elements from the Danek Bonebed, Upper Cretaceous Horseshoe Canyon Formation, Edmonton, Alberta, Canada: implications for allometric studies of fossil organisms. Canada Journal of Earth Science 51:1007–1016

Review form: Reviewer 2

Recommendation

Reject - article is scientifically unsound

Scientific importance: Is the manuscript an original and important contribution to its field?

Acceptable

General interest: Is the paper of sufficient general interest?

Acceptable

Quality of the paper: Is the overall quality of the paper suitable?

Acceptable

Is the length of the paper justified?

Yes

Should the paper be seen by a specialist statistical reviewer?

No

Do you have any concerns about statistical analyses in this paper? If so, please specify them explicitly in your report.

No

It is a condition of publication that authors make their supporting data, code and materials available - either as supplementary material or hosted in an external repository. Please rate, if applicable, the supporting data on the following criteria.

Is it accessible?

Yes

Is it clear?

No

Is it adequate?

Yes

Do you have any ethical concerns with this paper?

No

Comments to the Author

The manuscript by Knapp and colleagues tackles skull variation in *Protoceratops* using 3D geometric morphometrics (GM) to address the role of the skull's characteristic frill of this

ceratopsian dinosaur. I am aware of the controversy that took place a few years ago on the role of exaggerated structures in dinosaurs (cited in the literature and involving the authors), which, obviously, is why this paper hypothesizes that the frill has a socio-sexual signaling role. The study is well designed and it is well performed (see a few details on GM below) and also reads very well (there are a few loose sentences and missing commas here and there, but that is easily solvable and does not affect the quality of the paper). I must stress that I am not involved and I am completely neutral in this interpretative controversy, before I point out that, in my personal opinion, testing “roles” of traits – of whatever sort – in evolutionary biology is a bit farfetched. However, this is a philosophical position and shall not be my criticism. My real concern with this manuscript is that the results are obviously inconclusive regarding the hypotheses dealt with, and in turn, they can be interpreted in so many ways that directing interpretations to a particular “sexual” or “social” role of the frill seems somewhat forced. The problem is not that sexual dimorphic tests fail to address such a role to the frill, or that its variation is that of a module, hence behaving (variationally) selectively independent from other skull traits. The problem stems from the fact that a hypothesis as such (and an impacting result that should come thereof) would need to be carefully tested on an extremely controlled sample that represents a population of some sort (even if dealing with fossils), one that is sampled and studied in detail either in space (synchronous) or time (allochronous), or both. In the case of this study, for instance, a detailed mapping of precedence (strata) and regional-geographical relatedness of the studied fossils would be critically necessary (and evidence of embracing a relatively short time-span, required too). Of course, the fossils are in collections and it is what is available, yet a study aimed at resolving such a sound controversy would probably require field work or designing a way to address relatedness between the fossils in a reliable way. Without this, any hypothesis leaning on intraspecific biology will remain very difficult to prove. Otherwise, and let alone the problem of performing statistical inference with 30 specimens and more than 24k variables (i.e., coordinates from $p \approx 800$ landmarks/semilandmarks in 3D), the variation that was observed (and even the assessed modularity) might simply be interpreted as a phenomenon related to any other biological processes, adaptive (selected) or not (plasticity). The latter perhaps could be plausible, given the potential time-span between the strata where the specimens belonged to. Thus, as logic as the role of the frill might seem, this study’s arguments are weak enough to unnecessarily reignite the controversy on the role of exaggerated structures in dinosaurs, rather than solve it.

Particulars:

1. Graphically, the paper is good but I would exploit the graphic potential of 3D graphics (especially using so many landmarks and semilandmarks) with sequence deformations and 3D renders. The hard part was digitizing the coordinates, the rest is easy and relatively fast. Given that the authors are using over 800 points, using the whole skull’s data (1600) would do no extra harm to the stats, so why not render everything in more anatomically visual way? That’s one of GM potentials (and what is meant for). I would also include PCAs of aspects of variation that allow visualizing shape differences (ontogeny, for instance, which is what this paper ultimately deals with).
2. Modules were identified, yet it would be worth interesting any of them covaries in any way with any other module. There are several reasons for this. For instance, the frill must have been heavy, entailing that bones involved in muscle insertions and support might have changed “integratively” with it, at least in some degree (i.e., modules are semi-independent “units” that are loosely integrated with the rest). The fact that allometry underlies the patterns of variation of the modules indirectly suggests that the modules might be more integrated than expected.
3. I would avoid saying that one used X-command in R to perform an analysis (e.g., line 125 the “procD.lm”). That’s the equivalent of clicking on a button, and ultimately, via such command one uses a multivariate regression to test the covariation between size and shape (allometry). Likewise, the “diptest” is a rather intriguing test that I am not familiar with; more explanation

would be desirable.

Review form: Reviewer 3

Recommendation

Accept with minor revision (please list in comments)

Scientific importance: Is the manuscript an original and important contribution to its field?

Acceptable

General interest: Is the paper of sufficient general interest?

Good

Quality of the paper: Is the overall quality of the paper suitable?

Good

Is the length of the paper justified?

Yes

Should the paper be seen by a specialist statistical reviewer?

No

Do you have any concerns about statistical analyses in this paper? If so, please specify them explicitly in your report.

No

It is a condition of publication that authors make their supporting data, code and materials available - either as supplementary material or hosted in an external repository. Please rate, if applicable, the supporting data on the following criteria.

Is it accessible?

Yes

Is it clear?

Yes

Is it adequate?

Yes

Do you have any ethical concerns with this paper?

No

Comments to the Author

I thought this was a solid paper worthy of publication. That Protoceratops exhibits allometry of the frill in the absence of sexual dimorphism seems like a dead horse hardly worth beating anymore, but the insights into cranial modularity were novel. I have mostly minor concerns, spelled out below:

- Perhaps most egregiously, this paper fails to cite the recent work of Maiorino et al. (2015), who used very similar methods to reach very similar conclusions re: sexual dimorphism. This is a fairly massive oversight that needs to be corrected in the ms.

Maiorino, Leonardo, Andrew A. Farke, Tassos Kotsakis, and Paolo Piras. "Males resemble females: re-evaluating sexual dimorphism in *Protoceratops andrewsi* (Neoceratopsia, Protoceratopsidae)." *PloS one* 10, no. 5 (2015): e0126464.)

Despite the obvious overlap between these studies, I thought the current ms did a much better job of testing the modularity of the skull assumed by Maiorino et al., which sets this paper apart.

- Lines 38-45: Here, the authors first argue that sexual selection is a subset of social selection, but then (lines 42-43) say "the term 'socio-sexual selection' is thus used to refer to situations where either or both processes may operate." The word "either" seems to imply that sexual selection can occur in the absence of social selection, which does not jive with the authors' earlier comment that the former is just a subset of the latter. This phrasing needs attention.

- Line 121: I'm not sure what the "dilation component of taphonomic deformation" is. A sentence briefly explaining here would help.

- The authors used the dip test as a means of testing for sexual dimorphism. However, the dip test only tests for non-unimodality, which might be consistent with dimorphism, trimorphism, or any other number of -morphisms. It cannot by itself indicate dimorphism (which is why I also recommend use of mixture modeling). I don't think this affects the authors' results in any way, but it is a limitation that should be noted. It's probably also worth noting that the dip test is very conservative, as I say in my 2017 Paleobiology paper.

- Did the authors apply a correction to their multiple tests?

- Lines 169-170: What is it about the shape projection that suggests PC4 represents taphonomic deformation?

- Lines 251-254: A third possibility: The effect size of sexual dimorphism is small, and cannot be detected using the dip test.

Other comments are given in the ms markup.

Review form: Reviewer 4 (Caleb Brown)

Recommendation

Accept with minor revision (please list in comments)

Scientific importance: Is the manuscript an original and important contribution to its field?

Good

General interest: Is the paper of sufficient general interest?

Good

Quality of the paper: Is the overall quality of the paper suitable?

Excellent

Is the length of the paper justified?

Yes

Should the paper be seen by a specialist statistical reviewer?

Yes

Do you have any concerns about statistical analyses in this paper? If so, please specify them explicitly in your report.

No

It is a condition of publication that authors make their supporting data, code and materials available - either as supplementary material or hosted in an external repository. Please rate, if applicable, the supporting data on the following criteria.

Is it accessible?

Yes

Is it clear?

Yes

Is it adequate?

Yes

Do you have any ethical concerns with this paper?

No

Comments to the Author

Review of "Geometric morphometric analysis of the skull of *Protoceratops andrewsi* supports a socio-sexual signaling role for the ceratopsian frill"

The submission represents an important contribution to the ongoing debate regarding the evolutionary driver that results in the elaborate structures seen in many groups of dinosaurs. As such this paper is relevant to studies of dinosaur evolutionary patterns, as well as the morphological implications of socio-sexual selected structures in general, and in the fossil record specifically.

This submission attempts to test for four predictions of morphological structures that evolved under the influence of social-sexual selection, including positive allometry, high variation, modularity, and dimorphism. To test this, the authors use data of *Protoceratops*, a well-sampled taxon which is likely one of the best suited dinosaurs to test these ideas. The authors present evidence suggesting three of these predictions are met (though I would suggest these are stronger for some than others), and find little evidence for the fourth (dimorphism).

In general, the submission is a solid example of the potential investigations that can be undertaken with this type of sample, and represent the most robust and advanced treatment of this question in dinosaur palaeontology. The submission is well written and well figured, the analytical approach appropriate for testing the predictions laid out, and the interpretations supported by the results presented. The authors make all the primary data accessible (though see minor issues below), and should be commended on their well constructed and executed study. Despite this, there are some concerns that I have with the present submission. Some of these may represent issues that should be addressed prior to publication, while others may represent only minor concerns. Regardless, I do believe this is a submission suitable to this journal, and hope to see it published in the near future.

Caleb Brown

Major concerns

Dodson (1976): In many ways, this paper is an update of Peter Dodson's seminal 1976 work "Quantitative Aspects of Relative Growth and Sexual Dimorphism in *Protoceratops*". The taxon is the same. The dataset is similar (24 in 1976, 30 in 2020), and largely overlapping. And many of the predictions are the same: Dodson investigates patterns of allometry, variation and dimorphism (but not modularity), and finds evidence of strong positive allometry and variation in the frill (consistent with the present submission), but the results for dimorphism differ. The main difference are the analytical tools used. Dodson used linear morphometric (40 variables)

which was novel and ambitious at that time, whereas the current submission uses 3D geometric morphometrics (862 landmarks and semilandmarks) an equally ambitious approach for this time. However, despite this similarity, the work of Dodson 1976 is only mentioned cursorily in the submission. Once in the context (among other refs) that the frill has been suggested as a socio-sexual signal (line 97), once in the context of the nasal region possible functioning in combat (line 234), and once in the context of the ambiguity of the results for dimorphism (line 248). Never in the manuscript is it mentioned that Dodson (1976) produced broadly similar results to the present submission for the allometry and variation predictions (i.e., he showed strong positively allometry, and high variation in the frill), using much simpler methods 44 years ago. This is not to downplay the results of the current submission – which are important and should be published. But similarly, the submission should include a mention that Dodson was asking very similar questions of the same dataset in 1976, and indeed got very similar results (at least regarding allometry and variation). When the results of the current submission are reported, they should be contrasted with this previous work. It is interesting to note that this was done for the dimorphism section (where the results were not consistent), but was not done for the allometry and variation sections (where the results were broadly consistent, with the present work largely confirming the results of Dodson, 1976).

Locality of Specimens: There is no mention of the locality, both geographically and stratigraphically, from where these specimens were collected, nor is there a citation included that has these data. There is also no mention or citation about how the taxonomy of these specimens is determined. The methods section simply starts with the data collection of the landmarks. Are these specimens all recovered from the same locality? The same stratigraphic horizon? Do these represent a mass death assemblage of a herd, a penecontemporaneous sample of the population/species or are they a time averaged assemblage over tens/hundreds of thousands of years. Several recent papers have reported rapid morphological evolution of Ornithischian dinosaurs (including ceratopsians), specifically in their cranial ornamentation. I do not say this to imply that the results are invalid if this does not represent a single herd, or if these data are not known for each and every specimen. However, reporting these basic specimen level primary data are important for other researchers to evaluate the strength and weakness of the data presented in the submission.

There are examples of previous research examining variation amongst similar animals, for which stratigraphic data were not fully considered. These results were accepted as the best hypothesis for decades until the stratigraphic data were incorporated and the entire analysis fell apart.

Taphonomic Deformation: The submission suggests (line 109) “Specimens with small amounts of asymmetric taphonomic deformation were retrodeformed in Landmark v. 3.0 (32), using protocols outlined by Lautenschlager (33)” However, this is very short on details, and leaves many aspects unclear. Ideally, this should be detailed enough so that someone else can repeat the procedure and, at the moment, it is not.

How many specimens (and which) this was done for? This should be indicated.

How many landmarks/semilandmarks would be effected for this? Are these uniformly distributed or concentrated on thin/extreme areas like the frill?

I assume since the landmarking was done after the retrodeformation, the landmark data in the supplement are the adjusted data, not the raw data. Is the raw (i.e. pre retrodeformed) data available.

How was minor vs. severe taphonomic distortion determined?

What about symmetrically deformed specimens? Were these ignored?

Was there any test to see if retrodeformed specimens showed any distinct patterns from the non-deformed specimens in terms of variation, allometry etc.?

The submission states “The dilation component of taphonomic deformation, a potential source of non-biological variation, was identified and quantified by examining projected shapes created from PCA outputs of Procrustes-aligned data (40)” (line 120). This needs more detail in order to

be replicable. I am not sure if the data in the supplement include this analysis or not.

The submission states “Examination of the projected maximum and minimum shapes of the first 8 PCs suggest that PC4 (4.8% of total shape variation) represents the effects of taphonomic deformation (40)” (line 168). It is unclear how this was determined. How can taphonomic distortion be restricted to only one PC axis? I don’t think taphonomy can just be excluded so easily, is it in the following sentence.

The rostral landmarks of one specimen (AMNH 6409) were estimated using TPS interpolation. There is no indication why this estimation method was used. A recent paper (Arbour and Brown, 2014, MEE) has shown that TPS interpolation is consistently one of the poorest performing estimation methods and is often worse than simple mean substitution. It is also the only estimation method to perform worse than removing the incomplete specimens entirely (listwise deletion). TPS also tends to introduce the Pinocchio effect (disproportionally effecting peripheral landmarks) which will have implications for the fill of Protoceratops. I am not saying this method cannot be used, but at the very least the authors should justify why they chose this method of missing data estimation and the implications it may have on the dataset.

Dip Test Sample: The submission states “In addition to the 30 complete skulls used in the other analyses, 14 partial specimens were included in the dip test analysis of individual modules where the relevant modules were present in these specimens” (line 141). Which 14 specimens are these? Are their landmark data included in the supplement? Only data for 30 specimens are provided in the supplement.

Morphological variation: Of the three predictions that were tested with positive results, I find the results of the morphological variation least convincing. Looking at figure S12, the heatmap of variation across the 3D skull is basically a bulls-eye (see attached image). Low variation centrally (orbit/lacrymal) with a uniform increase in variation spreading out radially from the centre. This is most clear in B, where the variation increases radially towards the apices of the rostrum, jugal and posterolateral extreme of the frill. To me this is the exact pattern you would expect see as a result of plastic deformation of skulls due to taphonomy, where variation becomes concentrated at the thin and attenuated extremities, and is lowest at the central part of the skull. For these results to have more weight, there should be a discussion as to why this is not just the pattern expected for taphonomy or a radial artifact. Throughout the manuscript, I find the arguments for the dismissal of the effects of taphonomy unconvincing. I am inclined to believe the frill is more variable than the rest of the skull, but I do not find this figure to be convincing result of that pattern.

Smallest Specimens: The submission states “This was true for the dataset containing all complete specimens and for the dataset with smallest individuals removed” (196). There is no indication how the smallest specimens were selected for removal. How many were removed? Which ones? What size threshold was used? And how was this this size threshold determined? Without these data the analysis cannot be replicated.

Cast Specimens: Four of the specimens used in the analysis are casts. I have no methodological concerns with this (measurement error and small sample size will likely eclipse any issues with cast inaccuracy), but the original specimens that pertain to each cast need to be listed. These are the primary data.

Minor concerns:

Throughout the document, the sample size of the whole skull analysis is listed as 30 (lines 111, 141, 165, 191). In figure S8, however, the sample is listed as 31 in the caption. Only 30 points are visible, so I assume this is just a typo.

Throughout the document, the number of landmarks/semi-landmarks is recorded as 862 (line 862, figure S8 caption). However, in Supplementary Table S14, only 821 points are included. Which of these is correct, or did I miss something here.

Not a concern, just a question.. of the 6 modules, the postorbital shows the most divergent (and lowest) rate a size change though ontogeny (it is more divergent than the frill). I assume this is due to incorporating the orbit into its outline. To me this suggest that while the frill does shows divergent positive relative growth, the postorbital shows divergent negative relative growth – as would be expected. It this worth discussing?

Throughout the supplement the same numbering scheme is used for both the tables and figures, rather than each having their own. This seems odd, and I was confused thinking I had missing pages. Is this correct PRSB formatting? I would check and make sure these are numbered correctly.

Decision letter (RSPB-2020-1129.R0)

10-Jul-2020

Dear Mr Knapp:

I am writing to inform you that your manuscript RSPB-2020-1129 entitled "Geometric morphometric analysis of the skull of *Protoceratops andrewsi* supports a socio-sexual signalling role for the ceratopsian frill" has, in its current form, been rejected for publication in Proceedings B.

This action has been taken on the advice of referees, who have recommended that substantial revisions are necessary. With this in mind we would be happy to consider a resubmission, provided the comments of the referees are fully addressed. However please note that this is not a provisional acceptance.

Please note that this decision may (or may not) have taken into account confidential comments.

In your revision process, please take a second look at how open your science is; our policy is that *ALL* (maximally inclusive) data involved with the study should be made openly accessible, fully enabling re-use, replication and transparency-- see:
<https://royalsociety.org/journals/ethics-policies/data-sharing-mining/>

Insufficient sharing of data can delay or even cause rejection of a paper.
 Full data and code/scripts to enable reuse/replication/repurposing are what this policy intends.

Sincerely,
 Dr John Hutchinson, Editor
 mailto: proceedingsb@royalsociety.org

Associate Editor
 Board Member: 1
 Comments to Author:

Thank you for submitting your manuscript to Proceedings B. All reviewers agree this is an interesting contribution to the field and will appeal to the readership of PRSB. The reviewers do, however, raise several issues that they would like to see addressed before the manuscript is ready for publication. Most notably, a better grounding in the previous literature, more detail on the process of retrodeformation and further discussion on the stratigraphic location/temporal span of your sample.

Reviewer(s)' Comments to Author:

Referee: 1

Comments to the Author(s)

Knapp et al., Review Three-dimensional geometric morphometric analysis of the skull of *Protoceratops andrewsi* supports a socio-sexual signalling role for the ceratopsian frill
 The authors examined the skull of *Protoceratops andrewsi* using three-dimensional geometric morphometrics in order to assess whether the frill could be considered to have a role in socio-sexual signaling. To this end, they employed modularity analyses (EMMLi), tested for ontogenetic and static allometry, and examined morphological disparity of modules uncovered in their EMMLi analysis. They found that the *Protoceratops* skull is highly modular (8 modules) and found evidence that the frill evolved for socio-sexual signaling, but did not find evidence of sexual dimorphism. This is a strong paper and is well written. I particularly liked the structure of the paper (sets of predictions in the introduction mirrored in the discussion). The figures are also quite effective. I have outlined major comments and line-by-line comments below.

Comments:

I would like to see more of a discussion of taphonomic distortion. I have seen numerous *Protoceratops* skulls and none of them are completely without taphonomic distortion (many are quite distorted). The authors should include a discussion of the distortion in the discussion (likely between the penultimate and ultimate paragraphs). I understand that you did use the retrodeform function in landmark and then considered only 5% of total variation to be taphonomic with PC4 as your 'taphonomy PC'. However, there is not a single taphonomic distortion PC. All PCs will have some degree of taphonomic input. Hedrick et al. (2019) recently found that this can be as high as 40% even in relatively undeformed fossils. Tschopp et al. (2013) similarly found that distortion was extremely difficult to correct and the retrodeformation function in landmark makes the data more symmetric, but does not remove taphonomy from the equation. Hedrick and Dodson (2013) looked at 'relatively undeformed' psittacosaur skulls, but still found taphonomy to be a major driver in shape space. Not to mention that there is an unknowable amount of symmetric plastic deformation that may be affecting your skulls. This

certainly does not invalidate your analysis, but these issues must be discussed. Please expand on this. In addition to those above and Angielczyk and Sheets (2007), you can also look at Baert et al. (2014) and Arbour and Currie (2012).

More methodological details are needed. Piecing the methods and results together, I do not think that there are any issues. However, the methods are lacking in information. Here are a few examples: How was the photogrammetry performed? How were models decimated? How were semi-landmarks slid along surfaces (bending energy or Procrustes distance)? How was the dilation component of taphonomic deformation identified and quantified using PCA?

I would also like to see more of a discussion of sexual size dimorphism. Though osteological sexual shape dimorphism is sometimes found in extant taxa, sexual size dimorphism is much more common. Please expand this section, which may require additional analyses.

Please add an EMMLi table with all modularity hypotheses into the main text.

It would be great to expand on the discussion of modularity (lines 212-221) with extant examples. This is the first study of integration and modularity using GM in dinosaurs to my knowledge, but there is a rich extant literature. I think that expanding this part of the paper will make the paper more citable and expand its breadth to workers outside of paleobiology.

Line by line:

Line 19: What do you mean by 'high-dimensional'? 3D? Also line 102 and 284.

Line 20: Maybe 'Three predictions'? Three by itself reads strangely.

Line 25: By broader social contexts, do you mean species recognition?

Line 44: delete 'both'

Line 64: Modularity needs to be more clearly defined. It is not just that they are integrated sets of traits. Modules are highly integrated within themselves, but loosely integrated with adjacent modules. Please also add more citations to this. Although Klingenberg (2008) is one of the first papers to use modularity with GM, it is not the only important citation. Add Adams and Collyer (2016) and additional citations.

Line 69: 'as the basis for further'

Line 118: Are the results substantially different when you do not reflect landmarks? I have not seen this done before. Many datasets in the literature have GPA done on only one side.

Line 127: Modules are hypotheses and they have not yet been identified. Perhaps say hypothesized modules here?

Line 236: True; also, the nasal horn was likely covered in a keratinous sheath of unknown dimensions.

Figure 1: I find it a little confusing to have the dashed black line be isometry in 1A, but the trend line in 1B. Can you make the line in 1B a solid black line?

Adams, D.C. and M.L. Collyer. 2016. On the comparison of the strength of morphological integration across morphometric datasets. *Evolution*. 70:2623-2631.

Hedrick, B.P. and Dodson, P., 2013. Lujiatun psittacosaurids: understanding individual and taphonomic variation using 3D geometric morphometrics. *PLoS One*, 8(8).

Tschopp, E.D., Russo, J. and Dzemski, G., 2013. Retrodeformation as a test for the validity of phylogenetic characters: an example from diplodocid sauropod vertebrae.

Hedrick, B.P., Schachner, E.R., Rivera, G., Dodson, P. and Pierce, S.E., 2019. The effects of skeletal asymmetry on interpreting biologic variation and taphonomy in the fossil record. BRANDON P. HEDRICK ET AL. BIOLOGIC VARIATION IN THE FOSSIL RECORD. *Paleobiology*, 45(1), pp.154-166.

Arbour, V. M., and P. J. Currie. 2012. Analyzing taphonomic deformation of ankylosaur skulls using retrodeformation and finite element analysis. *PLoS ONE* 7:e39323.

Baert, M., M. E. Burns, and P. J. Currie. 2014. Quantitative diagenetic analyses of *Edmontosaurus regalis* (Dinosauria: Hadrosauridae) postcranial elements from the Danek Bonebed, Upper Cretaceous Horseshoe Canyon Formation, Edmonton, Alberta, Canada: implications for allometric studies of fossil organisms. *Canada Journal of Earth Science* 51:1007-1016

Referee: 2

Comments to the Author(s)

The manuscript by Knapp and colleagues tackles skull variation in Protoceratops using 3D geometric morphometrics (GM) to address the role of the skull's characteristic frill of this ceratopsian dinosaur. I am aware of the controversy that took place a few years ago on the role of exaggerated structures in dinosaurs (cited in the literature and involving the authors), which, obviously, is why this paper hypothesizes that the frill has a socio-sexual signaling role. The study is well designed and it is well performed (see a few details on GM below) and also reads very well (there are a few loose sentences and missing commas here and there, but that is easily solvable and does not affect the quality of the paper). I must stress that I am not involved and I am completely neutral in this interpretative controversy, before I point out that, in my personal opinion, testing "roles" of traits – of whatever sort – in evolutionary biology is a bit farfetched. However, this is a philosophical position and shall not be my criticism. My real concern with this manuscript is that the results are obviously inconclusive regarding the hypotheses dealt with, and in turn, they can be interpreted in so many ways that directing interpretations to a particular "sexual" or "social" role of the frill seems somewhat forced. The problem is not that sexual dimorphic tests fail to address such a role to the frill, or that its variation is that of a module, hence behaving (variationally) selectively independent from other skull traits. The problem stems from the fact that a hypothesis as such (and an impacting result that should come thereof) would need to be carefully tested on an extremely controlled sample that represents a population of some sort (even if dealing with fossils), one that is sampled and studied in detail either in space (synchronous) or time (allochronous), or both. In the case of this study, for instance, a detailed mapping of procedence (strata) and regional-geographical relatedness of the studied fossils would be critically necessary (and evidence of embracing a relatively short time-span, required too). Of course, the fossils are in collections and it is what is available, yet a study aimed at resolving such a sound controversy would probably require field work or designing a way to address relatedness between the fossils in a reliable way. Without this, any hypothesis leaning on intraspecific biology will remain very difficult to prove. Otherwise, and let alone the problem of performing statistical inference with 30 specimens and more than 24k variables (i.e., coordinates from p=ca. 800 landmarks/semilandmarks in 3D), the variation that was observed (and even the assessed modularity) might simply be interpreted as a phenomenon related to any other biological processes, adaptive (selected) or not (plasticity). The latter perhaps could be plausible, given the potential time-span between the strata where the specimens belonged to. Thus, as logic as the role of the frill might seem, this study's arguments are weak enough to unnecessarily reignite the controversy on the role of exaggerated structures in dinosaurs, rather than solve it.

Particulars:

1. Graphically, the paper is good but I would exploit the graphic potential of 3D graphics (especially using so many landmarks and semilandmarks) with sequence deformations and 3D renders. The hard part was digitizing the coordinates, the rest is easy and relatively fast. Given that the authors are using over 800 points, using the whole skull's data (1600) would do no extra harm to the stats, so why not render everything in more anatomically visual way? That's one of GM potentials (and what is meant for). I would also include PCAs of aspects of variation that allow visualizing shape differences (ontogeny, for instance, which is what this paper ultimately deals with).

2. Modules were identified, yet it would be worth interesting any of them covaries in any way with any other module. There are several reasons for this. For instance, the frill must have been heavy, entailing that bones involved in muscle insertions and support might have changed "integratively" with it, at least in some degree (i.e., modules are semi-independent "units" that are loosely integrated with the rest). The fact that allometry underlies the patterns of variation of the modules indirectly suggests that the modules might be more integrated than expected.

3. I would avoid saying that one used X-command in R to perform an analysis (e.g., line 125 the "procD.lm"). That's the equivalent of clicking on a button, and ultimately, via such command one uses a multivariate regression to test the covariation between size and shape (allometry). Likewise, the "diptest" is a rather intriguing test that I am not familiar with; more explanation would be desirable.

Referee: 3

Comments to the Author(s)

I thought this was a solid paper worthy of publication. That Protoceratops exhibits allometry of the frill in the absence of sexual dimorphism seems like a dead horse hardly worth beating anymore, but the insights into cranial modularity were novel. I have mostly minor concerns, spelled out below:

- Perhaps most egregiously, this paper fails to cite the recent work of Maiorino et al. (2015), who used very similar methods to reach very similar conclusions re: sexual dimorphism. This is a fairly massive oversight that needs to be corrected in the ms.

Maiorino, Leonardo, Andrew A. Farke, Tassos Kotsakis, and Paolo Piras. "Males resemble females: re-evaluating sexual dimorphism in *Protoceratops andrewsi* (Neoceratopsia, Protoceratopsidae)." *PloS one* 10, no. 5 (2015): e0126464.

Despite the obvious overlap between these studies, I thought the current ms did a much better job of testing the modularity of the skull assumed by Maiorino et al., which sets this paper apart.

- Lines 38-45: Here, the authors first argue that sexual selection is a subset of social selection, but then (lines 42-43) say "the term 'socio-sexual selection' is thus used to refer to situations where either or both processes may operate." The word "either" seems to imply that sexual selection can occur in the absence of social selection, which does not jive with the authors' earlier comment that the former is just a subset of the latter. This phrasing needs attention.

- Line 121: I'm not sure what the "dilation component of taphonomic deformation" is. A sentence briefly explaining here would help.

- The authors used the dip test as a means of testing for sexual dimorphism. However, the dip test only tests for non-unimodality, which might be consistent with dimorphism, trimorphism, or any other number of -morphisms. It cannot by itself indicate dimorphism (which is why I also

recommend use of mixture modeling). I don't think this affects the authors' results in any way, but it is a limitation that should be noted. It's probably also worth noting that the dip test is very conservative, as I say in my 2017 Paleobiology paper.

- Did the authors apply a correction to their multiple tests?

- Lines 169-170: What is it about the shape projection that suggests PC4 represents taphonomic deformation?

- Lines 251-254: A third possibility: The effect size of sexual dimorphism is small, and cannot be detected using the dip test.

Other comments are given in the ms markup.

Referee: 4

Comments to the Author(s)

Review of "Geometric morphometric analysis of the skull of *Protoceratops andrewsi* supports a socio-sexual signaling role for the ceratopsian frill"

The submission represents an important contribution to the ongoing debate regarding the evolutionary driver that results in the elaborate structures seen in many groups of dinosaurs. As such this paper is relevant to studies of dinosaur evolutionary patterns, as well as the morphological implications of socio-sexual selected structures in general, and in the fossil record specifically.

This submission attempts to test for four predictions of morphological structures that evolved under the influence of social-sexual selection, including positive allometry, high variation, modularity, and dimorphism. To test this, the authors use data of *Protoceratops*, a well-sampled taxon which is likely one of the best suited dinosaurs to test these ideas. The authors present evidence suggesting three of these predictions are met (though I would suggest these are stronger for some than others), and find little evidence for the fourth (dimorphism).

In general, the submission is a solid example of the potential investigations that can be undertaken with this type of sample, and represent the most robust and advanced treatment of this question in dinosaur palaeontology. The submission is well written and well figured, the analytical approach appropriate for testing the predictions laid out, and the interpretations supported by the results presented. The authors make all the primary data accessible (though see minor issues below), and should be commended on their well constructed and executed study. Despite this, there are some concerns that I have with the present submission. Some of these may represent issues that should be addressed prior to publication, while others may represent only minor concerns. Regardless, I do believe this is a submission suitable to this journal, and hope to see it published in the near future.

Caleb Brown

Major concerns

Dodson (1976): In many ways, this paper is an update of Peter Dodson's seminal 1976 work "Quantitative Aspects of Relative Growth and Sexual Dimorphism in *Protoceratops*". The taxon is the same. The dataset is similar (24 in 1976, 30 in 2020), and largely overlapping. And many of the predictions are the same: Dodson investigates patterns of allometry, variation and dimorphism (but not modularity), and finds evidence of strong positive allometry and variation in the frill (consistent with the present submission), but the results for dimorphism differ. The main difference are the analytical tools used. Dodson used linear morphometric (40 variables) which was novel and ambitious at that time, whereas the current submission uses 3D geometric morphometrics (862 landmarks and semilandmarks) an equally ambitious approach for this time.

However, despite this similarity, the work of Dodson 1976 is only mentioned cursorily in the submission. Once in the context (among other refs) that the frill has been suggested as a socio-sexual signal (line 97), once in the context of the nasal region possible functioning in combat (line 234), and once in the context of the ambiguity of the results for dimorphism (line 248). Never in the manuscript is it mentioned that Dodson (1976) produced broadly similar results to the present submission for the allometry and variation predictions (i.e., he showed strong positively allometry, and high variation in the frill), using much simpler methods 44 years ago. This is not to downplay the results of the current submission – which are important and should be published. But similarly, the submission should include a mention that Dodson was asking very similar questions of the same dataset in 1976, and indeed got very similar results (at least regarding allometry and variation). When the results of the current submission are reported, they should be contrasted with this previous work. It is interesting to note that this was done for the dimorphism section (where the results were not consistent), but was not done for the allometry and variation sections (where the results were broadly consistent, with the present work largely confirming the results of Dodson, 1976).

Locality of Specimens: There is no mention of the locality, both geographically and stratigraphically, from where these specimens were collected, nor is there a citation included that has these data. There is also no mention or citation about how the taxonomy of these specimens is determined. The methods section simply starts with the data collection of the landmarks. Are these specimens all recovered from the same locality? The same stratigraphic horizon? Do these represent a mass death assemblage of a herd, a penecontemporaneous sample of the population/species or are they a time averaged assemblage over tens/hundreds of thousands of years. Several recent papers have reported rapid morphological evolution of Ornithischian dinosaurs (including ceratopsians), specifically in their cranial ornamentation. I do not say this to imply that the results are invalid if this does not represent a single herd, or if these data are not known for each and every specimen. However, reporting these basic specimen level primary data are important for other researchers to evaluate the strength and weakness of the data presented in the submission.

There are examples of previous research examining variation amongst similar animals, for which stratigraphic data were not fully considered. These results were accepted as the best hypothesis for decades until the stratigraphic data were incorporated and the entire analysis fell apart.

Taphonomic Deformation: The submission suggests (line 109) “Specimens with small amounts of asymmetric taphonomic deformation were retrodeformed in Landmark v. 3.0 (32), using protocols outlined by Lautenschlager (33)” However, this is very short on details, and leaves many aspects unclear. Ideally, this should be detailed enough so that someone else can repeat the procedure and, at the moment, it is not.

How many specimens (and which) this was done for? This should be indicated.

How many landmarks/semilandmarks would be effected for this? Are these uniformly distributed or concentrated on thin/extreme areas like the frill?

I assume since the landmarking was done after the retrodeformation, the landmark data in the supplement are the adjusted data, not the raw data. Is the raw (i.e. pre retrodeformed) data available.

How was minor vs. severe taphonomic distortion determined?

What about symmetrically deformed specimens? Were these ignored?

Was there any test to see if retrodeformed specimens showed any distinct patterns from the non-deformed specimens in terms of variation, allometry etc.?

The submission states “The dilation component of taphonomic deformation, a potential source of non-biological variation, was identified and quantified by examining projected shapes created from PCA outputs of Procrustes-aligned data (40)” (line 120). This needs more detail in order to be replicable. I am not sure if the data in the supplement include this analysis or not.

The submission states “Examination of the projected maximum and minimum shapes of the first 8 PCs suggest that PC4 (4.8% of total shape variation) represents the effects of taphonomic deformation (40)” (line 168). It is unclear how this was determined. How can taphonomic distortion be restricted to only one PC axis? I don’t think taphonomy can just be excluded so easily, is it in the following sentence.

The rostral landmarks of one specimen (AMNH 6409) were estimated using TPS interpolation. There is no indication why this estimation method was used. A recent paper (Arbour and Brown, 2014, MEE) has shown that TPS interpolation is consistently one of the poorest performing estimation methods and is often worse than simple mean substitution. It is also the only estimation method to perform worse than removing the incomplete specimens entirely (listwise deletion). TPS also tends to introduce the Pinocchio effect (disproportionally effecting peripheral landmarks) which will have implications for the fill of Protoceratops. I am not saying this method cannot be used, but at the very least the authors should justify why they chose this method of missing data estimation and the implications it may have on the dataset.

Dip Test Sample: The submission states “In addition to the 30 complete skulls used in the other analyses, 14 partial specimens were included in the dip test analysis of individual modules where the relevant modules were present in these specimens” (line 141). Which 14 specimens are these? Are their landmark data included in the supplement? Only data for 30 specimens are provided in the supplement.

Morphological variation: Of the three predictions that were tested with positive results, I find the results of the morphological variation least convincing. Looking at figure S12, the heatmap of variation across the 3D skull is basically a bulls-eye (see attached image). Low variation centrally (orbit/lacrymal) with a uniform increase in variation spreading out radially from the centre. This is most clear in B, where the variation increases radially towards the apices of the rostrum, jugal and posterolateral extreme of the frill. To me this is the exact pattern you would expect see as a result of plastic deformation of skulls due to taphonomy, where variation becomes concentrated at the thin and attenuated extremities, and is lowest at the central part of the skull. For these results to have more weight, there should be a discussion as to why this is not just the pattern expected for taphonomy or a radial artifact. Throughout the manuscript, I find the arguments for the dismissal of the effects of taphonomy unconvincing. I am inclined to believe the frill is more variable than the rest of the skull, but I do not find this figure to be convincing result of that pattern.

Smallest Specimens: The submission states “This was true for the dataset containing all complete specimens and for the dataset with smallest individuals removed” (196). There is no indication how the smallest specimens were selected for removal. How many were removed? Which ones? What size threshold was used? And how was this this size threshold determined? Without these data the analysis cannot be replicated.

Cast Specimens: Four of the specimens used in the analysis are casts. I have no methodological concerns with this (measurement error and small sample size will likely eclipse any issues with cast inaccuracy), but the original specimens that pertain to each cast need to be listed. These are the primary data.

Minor concerns:

Throughout the document, the sample size of the whole skull analysis is listed as 30 (lines 111, 141, 165, 191). In figure S8, however, the sample is listed as 31 in the caption. Only 30 points are visible, so I assume this is just a typo.

Throughout the document, the number of landmarks/semi-landmarks is recorded as 862 (line 862, figure S8 caption). However, in Supplementary Table S14, only 821 points are included. Which of these is correct, or did I miss something here.

Not a concern, just a question.. of the 6 modules, the postorbital shows the most divergent (and lowest) rate a size change though ontogeny (it is more divergent than the frill). I assume this is due to incorporating the orbit into its outline. To me this suggest that while the frill does shows divergent positive relative growth, the postorbital shows divergent negative relative growth - as would be expected. It this worth discussing?

Throughout the supplement the same numbering scheme is used for both the tables and figures, rather than each having their own. This seems odd, and I was confused thinking I had missing pages. Is this correct PRSB formatting? I would check and make sure these are numbered correctly.

Author's Response to Decision Letter for (RSPB-2020-1129.R0)

See Appendix A.

RSPB-2020-2938.R0

Review form: Reviewer 1

Recommendation

Major revision is needed (please make suggestions in comments)

Scientific importance: Is the manuscript an original and important contribution to its field?

Acceptable

General interest: Is the paper of sufficient general interest?

Acceptable

Quality of the paper: Is the overall quality of the paper suitable?

Acceptable

Is the length of the paper justified?

Yes

Should the paper be seen by a specialist statistical reviewer?

No

Do you have any concerns about statistical analyses in this paper? If so, please specify them explicitly in your report.

No

It is a condition of publication that authors make their supporting data, code and materials available - either as supplementary material or hosted in an external repository. Please rate, if applicable, the supporting data on the following criteria.

Is it accessible?

Yes

Is it clear?

Yes

Is it adequate?

Yes

Do you have any ethical concerns with this paper?

No

Comments to the Author

I appreciate the author's changes based on my previous review. I find this paper to be quite interesting and a useful contribution for dinosaur biologists who use morphometrics or study ceratopsian taxa. I have one additional comment that I did not mention in the previous review and then a few minor comments (below).

I think that the paper would be made much stronger with clear extant examples that follow each of the author's four predictions for identifying socio-sexual signals. The fourth prediction, the presence of sexual dimorphism, is quite obviously related to sexual signaling and has been discussed extensively in the extant literature. However, each of the other predictions do not have as clear of a link to sexual selection and should be supported by findings in extants where the prediction has been shown to be linked to sexual signaling. Please also add alternative hypotheses surrounding each prediction.

For example in prediction 1, a module with low integration with adjacent modules is commonly considered to be highly evolvable, likely related to increased speciation and adaptive capacity, rather than sexual signaling. Mammals are a strong counter-example here because many mammalian species do have clear socio-sexual signaling, but the two-module cranial hypothesis appears to prevail across the majority of mammal clades. Predictions 3 and 4 similarly need more grounding in extant taxa. This will make the paper more relevant to a broader group of biologists.

Minor comments:

Line 27: ceratopsian should be lower-case

Line 126: Include Morpho package version number. Also for geomorph on line 131 and EMLLi on line 142.

Line 193: Do you mean the lowest change in shape with respect to size?

Line 239–40: I'm not sure what this means. Do you mean that *P. andrewsi* has a more integrated skull than archosaurs in general or other specific archosaur species?

Review form: Reviewer 2**Recommendation**

Reject – article is scientifically unsound

Scientific importance: Is the manuscript an original and important contribution to its field?

Acceptable

General interest: Is the paper of sufficient general interest?

Acceptable

Quality of the paper: Is the overall quality of the paper suitable?

Acceptable

Is the length of the paper justified?

Yes

Should the paper be seen by a specialist statistical reviewer?

No

Do you have any concerns about statistical analyses in this paper? If so, please specify them explicitly in your report.

No

It is a condition of publication that authors make their supporting data, code and materials available - either as supplementary material or hosted in an external repository. Please rate, if applicable, the supporting data on the following criteria.

Is it accessible?

Yes

Is it clear?

Yes

Is it adequate?

Yes

Do you have any ethical concerns with this paper?

No

Comments to the Author

The manuscript by Knapp and colleagues, tackling skull and frill variation in Protoceratops using 3D geometric morphometrics (Shape Analysis), addresses the biological/ethological role of the skull's characteristic frill of this ceratopsian dinosaur, as it did in its original form. The authors have paid considerable attention to reinforcing their statistical and graphic results, and the manuscript has clearly benefited from this. However, those weren't my deepest concerns with this study. Rather, such comments were intended to further underscore that, in my opinion, the results were inconclusive regarding the hypotheses dealt with. Unfortunately, as valuable as such results are, say, for descriptive purposes, the fact is that their big scope biological meaning has myriad possible interpretations, rather than the ones proposed. This shortcoming is not solved by placing null hypotheses as a point of departure, because such nulls are logic options, but not robust facts. Thus, although it is evident that sexual dimorphism tests fail to address such a role to the frill's shape variation, I find it intriguing that the authors remain concluding that their results "provide evidence that the exaggerated parietal-squamosal frills [...] fulfilled a socio-sexual signaling role". That the frill might have an intraspecific signaling role is so probable that testing it would seem unreasonable. Yet, scientific query is about demonstrating facts, hence the validity of this investigation. Unfortunately, though, I cannot help to see these results unconvincing on this matter. In my first review I argued that a detailed mapping of strata and regional-geographical relatedness of the studied fossils would be critical (and evidence of embracing a relatively short time-span, required too). The authors address the origins of the studied material and to me, as a paleontologist, it is obvious that they are right in that they are dealing with the same species. However, there is no sound reason to argue that the observed variation (and even the assessed modularity) might simply be interpreted as a phenomenon related to such biological processes, and not others, adaptive (selected) or not (particularly – and importantly – phenotypic plasticity). Furthermore, I tried to indicate that the latter (plasticity) perhaps would be more plausible for many paleobiologists (and neontologists), given the now-undisputedly demonstrated fact that phenotypic evolution is fast (environmentally cued) which is why this is the null hypothesis in microevolutionary research. And this is only an alternative,

given the radically new and unexpected facts that new fossil findings are revealing about dinosaur paleobiology.

This said, the descriptive results of this shape analytical study are definitely valuable to dinosaur paleobiologists, perhaps placed in a more specialized journal, and firmly proposing the hypotheses of the frill's ethological role in the discussion, and not as a point of departure.

Decision letter (RSPB-2020-2938.R0)

31-Dec-2020

Dear Mr Knapp

I am pleased to inform you that your manuscript RSPB-2020-2938 entitled "Geometric morphometric analysis of the skull of *Protoceratops andrewsi* supports a socio-sexual signalling role for the ceratopsian frill" has been accepted for publication in *Proceedings B*. Congratulations!!

The referee(s) have recommended publication, but also suggest some minor revisions to your manuscript. Therefore, I invite you to respond to the referee(s)' comments and revise your manuscript. Because the schedule for publication is very tight, it is a condition of publication that you submit the revised version of your manuscript within 7 days. If you do not think you will be able to meet this date please let us know.

While we call these revisions "minor", the 2 reviewers agree that there are serious issues with how the hypotheses of signalling have been addressed vs. alternatives, and these need to be better addressed in revisions, with citations of evidence from extant taxa and defense vs. the phenotypic plasticity concern wherever feasible. However, as this MS has already gone through extensive review and some prior reviews were very positive about it, we find the study overall to be acceptable with such revisions, which would make it more broadly relevant and robust.

- 1) A text file of the manuscript (doc, txt, rtf or tex), including the references, tables (including captions) and figure captions. Please remove any tracked changes from the text before submission. PDF files are not an accepted format for the "Main Document".
- 2) A separate electronic file of each figure (tiff, EPS or print-quality PDF preferred). The format should be produced directly from original creation package, or original software format. PowerPoint files are not accepted.

3) Electronic supplementary material: this should be contained in a separate file and where possible, all ESM should be combined into a single file. All supplementary materials accompanying an accepted article will be treated as in their final form. They will be published alongside the paper on the journal website and posted on the online figshare repository. Files on figshare will be made available approximately one week before the accompanying article so that the supplementary material can be attributed a unique DOI.

Sincerely,

Dr John Hutchinson, Editor

Associate Editor

Board Member

Comments to Author:

Thank you for resubmitting your manuscript to Proceedings B. We appreciate the time and effort the authors have spent in addressing the concerns of the reviewers. Reviewer 1 would now like to see some extant examples to provide further support to the authors' 4 study predictions. I would strongly encourage the authors to incorporate such examples, as it will likely broaden the appeal of the manuscript to a more diverse suite of readers. Reviewer 2 still has concerns regarding the inconclusive nature of the results. Where possible, I would encourage the authors to include a discussion/caveat regarding the potential for the observed morphological variation to be somewhat a function of phenotypic plasticity as opposed to other adaptive processes like socio-sexual signalling.

Reviewer(s)' Comments to Author:

Referee: 1

Comments to the Author(s).

I appreciate the author's changes based on my previous review. I find this paper to be quite interesting and a useful contribution for dinosaur biologists who use morphometrics or study ceratopsian taxa. I have one additional comment that I did not mention in the previous review and then a few minor comments (below).

I think that the paper would be made much stronger with clear extant examples that follow each of the author's four predictions for identifying socio-sexual signals. The fourth prediction, the presence of sexual dimorphism, is quite obviously related to sexual signaling and has been discussed extensively in the extant literature. However, each of the other predictions do not have as clear of a link to sexual selection and should be supported by findings in extants where the prediction has been shown to be linked to sexual signaling. Please also add alternative hypotheses surrounding each prediction.

For example in prediction 1, a module with low integration with adjacent modules is commonly considered to be highly evolvable, likely related to increased speciation and adaptive capacity, rather than sexual signaling. Mammals are a strong counter-example here because many mammalian species do have clear socio-sexual signaling, but the two-module cranial hypothesis appears to prevail across the majority of mammal clades. Predictions 3 and 4 similarly need more grounding in extant taxa. This will make the paper more relevant to a broader group of biologists.

Minor comments:

Line 27: ceratopsian should be lower-case

Line 126: Include Morpho package version number. Also for geomorph on line 131 and EMLLi on line 142.

Line 193: Do you mean the lowest change in shape with respect to size?

Line 239–40: I'm not sure what this means. Do you mean that *P. andrewsi* has a more integrated skull than archosaurs in general or other specific archosaur species?

Referee: 2

Comments to the Author(s).

The manuscript by Knapp and colleagues, tackling skull and frill variation in Protoceratops using 3D geometric morphometrics (Shape Analysis), addresses the biological/ethological role of the skull's characteristic frill of this ceratopsian dinosaur, as it did in its original form. The authors have paid considerable attention to reinforcing their statistical and graphic results, and the manuscript has clearly benefited from this. However, those weren't my deepest concerns with

this study. Rather, such comments were intended to further underscore that, in my opinion, the results were inconclusive regarding the hypotheses dealt with. Unfortunately, as valuable as such results are, say, for descriptive purposes, the fact is that their big scope biological meaning has myriad possible interpretations, rather than the ones proposed. This shortcoming is not solved by placing null hypotheses as a point of departure, because such nulls are logic options, but not robust facts. Thus, although it is evident that sexual dimorphism tests fail to address such a role to the frill's shape variation, I find it intriguing that the authors remain concluding that their results "provide evidence that the exaggerated parietal-squamosal frills [...] fulfilled a socio-sexual signaling role". That the frill might have an intraspecific signaling role is so probable that testing it would seem unreasonable. Yet, scientific query is about demonstrating facts, hence the validity of this investigation. Unfortunately, though, I cannot help to see these results unconvincing on this matter. In my first review I argued that a detailed mapping of strata and regional-geographical relatedness of the studied fossils would be critical (and evidence of embracing a relatively short time-span, required too). The authors address the origins of the studied material and to me, as a paleontologist, it is obvious that they are right in that they are dealing with the same species. However, there is no sound reason to argue that the observed variation (and even the assessed modularity) might simply be interpreted as a phenomenon related to such biological processes, and not others, adaptive (selected) or not (particularly – and importantly – phenotypic plasticity). Furthermore, I tried to indicate that the latter (plasticity) perhaps would be more plausible for many paleobiologists (and neontologists), given the now-undisputedly demonstrated fact that phenotypic evolution is fast (environmentally cued) which is why this is the null hypothesis in microevolutionary research. And this is only an alternative, given the radically new and unexpected facts that new fossil findings are revealing about dinosaur paleobiology.

This said, the descriptive results of this shape analytical study are definitely valuable to dinosaur paleobiologists, perhaps placed in a more specialized journal, and firmly proposing the hypotheses of the frill's ethological role in the discussion, and not as a point of departure.

Author's Response to Decision Letter for (RSPB-2020-2938.R0)

See Appendix B.

Decision letter (RSPB-2020-2938.R1)

12-Jan-2021

Dear Mr Knapp

I am pleased to inform you that your manuscript entitled "Geometric morphometric analysis of the skull of *Protoceratops andrewsi* supports a socio-sexual signalling role for the ceratopsian frill" has been accepted for publication in *Proceedings B*. Congratulations!!

Open Access

Your article has been estimated as being 9 pages long. Our Production Office will be able to confirm the exact length at proof stage.

Paper charges

Sincerely,

Dr John Hutchinson

Appendix A

Response to referees

Referee 1

Taphonomic deformation: A more detailed discussion of taphonomic deformation has been added to the discussion (lines 280 – 290). We agree with the referee that our estimate of 5% contribution of taphonomic deformation to total shape variation is likely an underestimate. We have reworded our discussion of non-allometric shape variation to reflect this and to be more cautious in our interpretation of this variation (lines 190 – 193).

Methodological detail: More details have been added to the methods section explaining photogrammetry (lines 116 – 117), model construction (line 118), semilandmark sliding (line 129) and estimating deformation from PCs (lines 136 – 139).

Sexual size dimorphism: Previous studies (e.g. Dodson, 1976, Maiorino et al., 2015, and Mallon, 2017) have attempted to detect sexual size dimorphism in *P. andrewsi* using linear measurements. Hone and Mallon (2017) showed that dimorphism is hard to detect without ontogenetic age correction and large numbers of specimens (70+ in their analysis) using linear measurements. Maiorino et al. (2015) also applied 2D morphometrics to tackle the question of sexual dimorphism in this taxon. Because of the number of previous studies examining size dimorphism in *P. andrewsi* we focussed our study on the analysis of shape, attempting to detect dimorphism with high-density 3D geometric morphometrics. Size dimorphism was, therefore, not a product of our study. We have modified some text in the discussion to clarify this (lines 305 – 323).

EMMLi table: We have included a full modularity table in the supplementary file (Supplementary Table 5), covering the EMMLi and covariance ratio (CR) outputs for both raw and allometry-corrected data. We felt that the limited space of the article would be better served by the included 3 figures, although we would be happy to include the table with the main text if the editor considers it a vital contribution.

Modularity: We have added a paragraph discussing the ontogenetic modularity of *P. andrewsi* and evolutionary modularity that has been previously explored in a number of extant and extinct clades (lines 243 – 253).

Line by line (updated to reflect current line numbers).

Line 19: ‘High-dimensional’ refers to the situation where numbers of landmarks are higher than the number of specimens, i.e. a high—density landmark approach for capturing shape. This has been changed to high-density for clarity.

Line 20: Corrected to ‘four predictions’.

Line 26: ‘Broader social contexts’ referred to more general intraspecific interactions. This has been corrected to ‘more general social interactions.’

Line 44: ‘both’ deleted.

Line 63: The reviewer is correct that modularity refers to sets of traits that show relatively high values of internal integration values compared with integration values between other modules. The wording has been clarified to reflect this, and additional references added.

Line 69: ‘as the basis for further’ wording corrected.

Line 129: The skulls of tetrapods are generally, with some exceptions (e.g., odontocetes), symmetrical. When performing a Procrustes alignment, significant variation in width of the skull (as seen during

ontogeny in the frill of *Protoceratops*) acts to pull narrower specimens towards the lateral extremities of wider specimens and shift the centroid away from the sagittal plane, affecting alignment accuracy (see Cardini, 2016). The sagittal plane therefore acts as a fixed reference for skull alignment in this situation. It is unlikely that this will affect the results greatly, but will tend to increase landmark variance towards the midline of the skull/sagittal plane and reduce it towards the lateral extremities (a variation on the Pinocchio effect). The wording in the methods section has been edited to explain more clearly the reasons for doing this.

Line 141: This line was a little unclear; although modules had not been identified at this stage the method was also used to quantify individual module shape change at a later stage, when they had. The wording has been changed to “The ‘common allometric component’ (CAC) is the component of shape variation most closely aligned with size.” because it can be applied to all sets of shape data.

Line 270: Agreed, a line about keratinous sheaths being associated with horns has been added.

Figure 1: As recommended, the dashed line representing whole-skull shape regression against size has been changed to a solid line for consistency with the regression lines for individual modules.

Referee 2

The reviewer raises an important point about the origin of fossil specimens and how this may affect variation within the dataset. Many specimens of *P. andrewsi* were collected in the early 20th century and no detailed stratigraphic data exists for these specimens. The same is, unfortunately, also true for many more recently-collected specimens. All of the specimens included in this study, however, were collected from the Djadochta formation of Mongolia and were found in just two locations, Bayan Dzak and Tugrugyin Shireh. Evidence suggests that the units at these two locations that yield specimens of *Protoceratops andrewsi* are of equal or nearly equal age (Czepinski, 2020). The members from which the specimens are found cover a fairly restricted area at the two sites, and are from single fossil-bearing layers that range in thickness from 2 to 14 metres (Dashzeveg et al., 2005). The restricted localities from which the specimens in our study were collected, coupled with diagnostic features of *P. andrewsi* (Presence of paired nasals, the premaxillary dentition, the absence of the accessory antorbital fenestration and U-shaped buccal crest of the dentary; Czepinski, 2020) have led the majority of researchers to agree that *P. andrewsi* is the only ceratopsian taxon found at these locations (Czepinski, 2020). Indeed, the majority of specimens we have included in our analysis have featured in a number of previous intraspecific analyses of *P. andrewsi*. The focus of our study is a fossil taxon and, as the referee notes, we can only work with the specimens we have. Even so, *Protoceratops* represents one of the best taxa available that is not known from a single mass assemblage (which are often represented by fragmentary remains). We have included a discussion on the origin of the specimens in our study in the supplementary file covering these points, and also performed additional analyses on the shape data in the form of a MANOVA to detect shape differences between specimens from the two locations. No significant differences were found. Nevertheless, we reran the main analyses on all specimens from each locality. The results of these analyses, which do not affect the findings of our study, have been summarised in the supplementary information (Supplementary Tables 3 and 8).

The referee’s point about making statistical inference from high-dimensional data such as ours is an important one, and is much discussed in geometric morphometric literature. The issues and solutions surrounding it are well-summarised in Goswami et al., 2019. The problem is that anatomical landmark-only datasets do a poor job of capturing shape variation, particularly in regions with few easily—

identifiable homologous points (e.g. the curved rear edge of the frill in our dataset), and increasing the number of semilandmarks is often unavoidable when dealing with more complex morphologies. Additionally, modularity analyses, which require measures of covariation both within and between regions, tend to show higher inter- and lower within-region integration when regions are described by far fewer landmarks. The multivariate methods we have used in our analyses have been developed for dealing with high-dimensional data such as ours, and have been employed in a number of previous studies. Furthermore, a number of our analyses, such as the allometry analyses in Fig. 1, are based on the outputs of PCAs which reduce dimensionality to well below the number of specimens in the study and so are compatible with more simpler statistical methods such as ANOVAs to compare allometric slopes Goswami et al. (2019) demonstrate in a number of empirical and simulated datasets that 'increasing landmark or semilandmark sampling alone does not exacerbate issues with procedures such as Procrustes analysis'. To demonstrate that our dataset is robust to the dimensionality, we reran a number of analyses using only the anatomical landmarks in our dataset (n=28 landmarks). These landmarks are shown as red points in Supplementary Fig. 2, and are the first 28 points listed in Supplementary Table 2. The results of these analyses are summarised in Supplementary Table 8. Procrustes variance is much reduced in these analyses, as we might expect given the omission of a great deal of shape information. Modularity patterns derived from EMMLi are difficult to interpret due to the issues noted above, but covariance ratio analysis does, however, indicate a significantly modular structure ($p = 0.02$). The results of the PCA of this subsampled dataset are very similar to that of the full dataset (PC1 = 64.0% of shape variation in reduced dataset cf. 63.1% in full dataset). Similarly, the allometry analysis yields near identical results ($R^2 = 0.51$, $p = 0.004$ reduced, $R^2 = 0.52$, $p = 0.004$ full). These additional analyses broadly agree with the assessment of Goswami et al. (2019) and show that even when greatly reduced, our conclusions are largely robust to the number of landmarks.

Particulars:

1. We have included a number of graphics that illustrate changes in morphology across several principal components (including that of the allometric trajectory, see Figure 3), and the change in shape of individual modules through ontogeny (Fig. 2). Due to space limitations, others are found in the supplementary data (Supplementary Figs. 4 and 6), including additional figures in the supplementary data to show shape changes associated with more principal components.
2. Between-module correlation values are included in the modularity tables in the supplementary figures (Supplementary Table 5), including a network plot to illustrate the strength of these interactions. Again, due to space limitations these were not included in the main manuscript but are referenced where appropriate.
3. We agree with reviewer 2 on the point that the emphasis should be on explaining the statistical method involved rather than the R function used; statistical methods have been more explicitly explained in the manuscript (see lines 140, 144 and 163).

Referee 3

We are of course familiar with the work of Maiorino et al. (2015), and it appears in earlier versions of our manuscript. Its omission in the submitted manuscript we can only attribute to overenthusiastic editing. This has been corrected (lines 106, 235, 255, 295, 296).

As an additional comment, we agree with referee 3 that there is obvious overlap between Maiorino et al. (2015) and our study, and think that we needed to emphasise that our study moves from 2D to 3D data (line 109), incorporates a number of new specimens (line 107, Supplementary Table 1), and also employs a number of different analyses (line 235).

Individual lines (line numbers updated to current manuscript)

Lines 39-44: We agree that our original wording concerning social and sexual selection was ambiguous, it has been clarified.

Line 135: A brief explanation of the dilation component of taphonomic deformation has been added.

We agree with the referee's comment about the dip test. We did not expect to find any clear evidence of dimorphism in the dataset, given previous studies and examination of our data, but used this as an opportunity to apply it to shape (rather than size) data for the first time. Our reasoning has been explained in the manuscript (lines 160-163).

"Did the authors apply a correction to their multiple tests" We are not sure which 'multiple tests' the reviewer was referring to here, All modularity, morphological disparity and dip tests were performed with raw and allometry-corrected data, and all analyses were rerun with certain specimens omitted to ensure that possible errors are accounted for (results of additional analyses are included in Supplementary Table 8).

Lines 189-192: The projected shapes of PC4 show a change in shape that is consistent with a small amount of lateral to dorso-ventral compression (although this is not as obvious in any individual specimen). This has been clarified in the manuscript, and additional principal component projected shapes have been added to the supplementary file (Supplementary Fig. 6) to better illustrate this.

Line 297 – 299: In our example small effects of sexual shape dimorphism is covered by similar shape trajectories between sexes. This has been clarified in the text.

Referee 4

Major concerns

Dodson (1976) was the first major quantitative study to make use of the large number of *Protoceratops* specimens in the AMNH, and, as the referee points out, the aims of our study and those of Dodson have some overlap. Although Dodson lists 24 specimens in his original study this number included partial specimens. The total number of complete specimens in Dodson's 1976 study was 13, whereas our study uses 30 complete specimens (plus an additional 14 partial specimens for the dimorphism analysis). This is because we were able to access a number of specimens in the Mongolian Academy of Sciences in Ulaanbaatar which were unavailable to Dodson. The details of all specimens, including origin, have been expanded upon in the supplementary file (Supplementary Table 1). Although Dodson did measure allometry in his 1976 paper, he did not statistically compare allometry slopes, as did Hone et al. (2016) and O'Brien et al. (2018). We used module centroid size to compare all modules, but also compared shape changes through ontogeny. This has been previously used with primates (Mitteroecker et al., 2004), fish (Evans et al., 2018), and to compare frill shape changes across ceratopsian taxa (Prieto-Marquez et al., 2020), but here we use it to compare shape changes of skull modules during ontogeny for the first time in a dinosaur taxon. Nonetheless, because of the importance of Dodson's work, we have expanded the comparison of our results and Dodson's (lines 102, 106, 262, 280, 295, 296).

Specimen locality: As we have responded to referee 2 above, the origin of the specimens in our study has been added to the specimen list in the supplementary files (Supplementary Table 1). Because of the lack of precise stratigraphic data for the majority of Mongolian fossils, we cannot know with the same accuracy as a more well-studied location as, for example, Dinosaur Provincial Park, but the much more limited extent of the exposure from which these fossils are found (Dashzeveg et al., 2005), along with a number of diagnostic traits (Czepinski, 2019), can give us some degree of confidence that the sample we have represents a valid group for intraspecific analysis (Czepinski, 2020).

Taphonomic deformation: The method we used to identify specimens for retrodeformation has been added to the manuscript (lines 117 – 120), and specimens that were treated in this way have been identified in the supplementary specimen data (Supplementary Table 1). In addition, the retrodeformed specimens were landmarked and reanalysed in their original, un-retrodeformed shape (Supplementary Table 8). This raw landmark data has been included as a supplementary file, along with landmark data for individual modules as employed in the dip tests.

As mentioned to referee 3, the projected maximum and minimum shapes of the first four principal components have been added to the supplementary data (Supplementary Fig. 6). After consideration of our originally-submitted discussion, we agree that taphonomic deformation should be discussed in more detail. Accordingly, we have expanded our discussion on the subject (lines 279 – 289).

We understand the concern with estimating missing landmark data. This process was used only on one specimen because it was complete other than the missing rostral bone, the smallest component of the skull in the analysis and consisting of only one anatomical landmark and 16 semilandmarks (out of a total of 821 landmarks for the entire skull). No part of the main anatomical focus of this study (i.e. the frill) was estimated. We felt that this was the best approach because, as mentioned in Arbour and Brown (2014), it mitigated the loss of data from excluding an otherwise complete specimen. We used the TPS method because of its suitability to our dataset; we estimated a relatively small number of landmarks from a small portion of the skull of a densely-landmarked, single-taxon dataset. Arbour and Brown (2014) suggest that the TPS method would be more accurate under these dataset conditions. To ensure that our final analyses were not affected by this, we ran the analyses with this specimen excluded and found no appreciable difference in the results (Supplementary Table 8).

Dip test sample: Landmark data for each individual module analysed with the dip test, including that for incomplete specimens, have been included as separate .csv files for each module.

Morphological variation: We agree with referee 4 that Supplementary Figure S12 in our original submission is unclear. Originally, two images (one of total per-landmark variance and one of allometry-corrected per-landmark variance) were shown side-by-side, but the Procrustes variance values for each were individually scaled, giving the impression that total variation in the corrected dataset was as high as for the total dataset. In our updated figure (Supplementary Fig. 9) we have scaled the variance values relatively to give a better impression of the effect of allometry on variance compared with variance due to other factors. We calculated the log of the variance values to show a smoother transition between low and high variance regions. This has the effect of increasing the colour value range of lower variance landmarks, and may give the impression that some regions have relatively higher variance. Conversely, it also allows us to show more subtle differences in variance between regions. Because this figure is mainly for illustrative reasons, we opted to show the log values to better describe smaller-scale variances. Plots of individual landmarks variance on an XY plot have also been included in this figure for reference.

The 'bullseye' effect in our original Supplementary Figure S12 noted by referee 4 seems to be an artefact of the orientation of the original figure. The region of lowest variance is centred in the image and the region of highest variance (i.e. the lateral extremities of the frill) are towards the perimeter. While we accept that this pattern may result from taphonomic deformation of the skull, we do not think that it is the sole cause of the pattern we see in this figure. Firstly, the high variance seen at the border of the frill corresponds with the extreme expansion and highly significant shape/size relationship of this structure during ontogeny (Figure 3). Secondly, although it seems that variance shows a smooth transition across the skull from the low variance 'bullseye' to the frill, closer examination of the allometry-corrected figure shows that this is not exactly the case; regions of the frill, particularly the portion of the parietal anterior to the parietal fenestra shows among the very lowest variance values of the entire skull and is farther from the region of low variance forward of the orbit than the squamosal/postorbital suture, which itself shows comparatively higher variance. We think that the extreme thinness of the parietal in this region would show higher variance values if taphonomic deformation was the major cause of variation in the skull, and particularly the frill. Additionally, the rearmost region of the frill on the centre line may be expected to show similar variance values to the lateral extremities if this variance was due entirely to deformation by compression, because plastic deformation (i.e. deformation not involving fractures) of part of the skull would inevitably result in changes elsewhere due to conservation of volume. Furthermore, we do not see correspondingly high variance values in the orbit, which is known to be particularly prone to taphonomic deformation (Arbour and Currie, 2012; Kammerer et al., 2020). Finally, unless taphonomic deformation affected all specimens in the same way, we would not expect a 'centre' of deformation creating a bullseye-like pattern in a specific region of the skull away from the midline. This is because specimens are Procrustes-aligned along this axis, and we would therefore expect variance to be lowest at this point as deformed regions are pushed towards or away from it (there is no asymmetrical component of deformation in our analysis because we analysed only one side of the skull). Inevitably, some variation in our dataset is due to taphonomic deformation, and we concede that determining the exact amount is difficult. We have included shape projections of the first 4 PCs (Supplementary Fig. 6) that we used to estimate the contribution of taphonomic deformation to shape variation, but with the exception of PC4, there is no obvious pattern of deformation in the remaining PCs (e.g. dorsal compression corresponding with lateral dilation, and vice versa). It is, however, clear from our analyses that the overwhelming source of shape variation in our dataset is significantly correlated with size, and deformation does not appear to overwhelm biological signal, and we have included more on this subject in the discussion (lines 279-289).

Smallest Specimens: The smallest specimens were determined by examining the output of the allometry analysis. Two distinct clusters representing the smallest specimens can be identified in the figure (Supplementary Fig. 7), with the larger 'adult' specimens forming a continuous group and separated from the 'juveniles' by the mean value. The process for identifying these specimens has been added to the manuscript (line 171), specimen groups have been indicated in Supplementary Fig. 7, and specimens designated as 'juvenile' have been indicated in the Supplementary Table 1.

Cast specimens: Where possible, the original specimens on which casts are based have been listed in Supplementary Table 1. In the case of two specimens (QMBC 1293 and 1294) the originals are not known. These two casts show all of the diagnostic features consistent with *P. andrewsi*, and both fall within the region of morphospace occupied by the known specimens. The lack of locational information means, however, that we cannot be certain that they are from the same two localities as the remaining specimens. To account for this we re-ran the modularity, allometry, variance and dip test analyses with these two specimens omitted (Supplementary Table 8). The omission of these specimens did not appreciably affect the results in any analysis.

Minor concerns:

The number of specimens listed in Supplementary Figure 4 is indeed a typo. It has been corrected.

The total number of landmarks and semilandmarks in the final analysis is 821. This seems to have been a duplicated error based on the number of landmarks in an earlier landmarking scheme, which has been corrected in the manuscript.

We're not sure we would describe the postorbital as the most divergent module, rather that it shows the lowest relative size change when compared to the entire skull; other modules change much more in size (but not necessarily in shape) through ontogeny. To truly answer this question, we would need to compare it to a number of other ceratopsian taxa, but we do not really have sufficient data for this at present. We agree that this is likely due to incorporating the majority of the orbit margin into this module, and reflects the negative allometry generally seen in orbit size across a range of taxa, but we do not think it is worth discussing this in detail because of space limitations and the focus of our study being that of the frill.

Proceedings B does not have a specific formatting style for the supplement. To avoid confusion, we have changed the numbering and order of the supplementary figures and tables.

Additional references

Arbour VM, and Currie PJ (2012). Analyzing taphonomic deformation of ankylosaur skulls using retrodeformation and finite element analysis. *PLoS ONE*. **7(6)**: e39323. doi:10.1371/journal.pone.0039323.

Arbour JH and Brown CM (2014). Incomplete specimens in geometric morphometric analyses. *Methods in Ecology and Evolution*. **5(1)**: 16-26.

Czepiński Ł (2019). New protoceratopsid specimens improve the age correlation of the Upper Cretaceous Gobi Desert strata. *Acta Palaeontol. Pol.* **65**. Doi: 10.4202/app.00701.2019.

Dashzeveg D, L Dingus DB, Loope CC, Dulam ST, and Sweeney MR (2005). New stratigraphic subdivision, depositional environment, and age estimate for the Upper Cretaceous Djadokhta Formation, Southern Ulan Nur Basin, Mongolia. *American Museum Novitates* **3498**:1–31.

Goswami A, Watanabe A, Felice RN, Bardua C, Fabre AC and Polly PD (2019). High-density morphometric analysis of shape and integration: the good, the bad, and the not-really-a-problem. *Integrative and Comparative Biology*. <https://doi.org/10.1093/icb/icz120>..

Appendix B

Response to referees

Dear editors

We are delighted that you have decided to accept our manuscript. We thank the reviewers again for their thoughtful comments and suggestions. In line with your recommendation, we have made some additions to the manuscript as requested by the reviewers.

Referee 1

As requested, we have added some specific examples to our predictions in the introduction. One prediction where this is not especially straightforward is that of modularity. Modularity is expected to occur in instances where traits become integrated and are able to respond to selection somewhat independently of others. This should be true of any adaptive set of traits, including those used for socio-sexual selection (e.g. display traits, weapons etc.). Of course, forming a phenotypic module is not unique of socio-sexual traits, but it does allow us to justify our partitioning the skull into separate, integrated units which we may statistically compare. The majority of detailed cranial modularity analyses to date have focussed on large-scale evolutionary patterns, and these tend to avoid including taxa with exaggerated cranial ornaments because this may mask more subtle (and evolutionarily informative) underlying morphology. An example is the 2017 study by Felice and Goswami, looking at cranial integration in the avian skull. In sexually dimorphic taxa, the sex with smaller or absent ornaments was used to minimise this effect. Nevertheless, one notable modularity study (Watanabe et al., 2019) has found a link between squamate cranial modules that showed rapid morphological evolution and high disparity, and cranial ornamentation. We have included this example in our revised manuscript, and expanded our explanation.

There are numerous examples of secondary sexual signals which display positive allometry in extant taxa (though not all secondary sexual traits do), and we have included several examples from taxonomically diverse groups. Although positive allometry does occur in non-sexual traits, it is often not significantly higher than either isometry or other positively-scaling traits within the same organism. An exception appears to be locomotory traits in some taxa, where positive scaling is thought to provide benefits in movement efficiency. We have included a known example of this. In addition, we have included examples of extant taxa in which secondary sexual traits show markedly higher variation than non-sexual 'reference' traits.

Minor comments:

Line 27: ceratopsian changed to lower case

Line 126: Morpho package number added

Line 193: This has been corrected to 'lowest change in module size with respect to skull size'

Line 239-240: *P. andrewsi* has a more integrated skull than archosaurs in general. This has been clarified.

Referee 2

We have discussed the phenotypic plasticity concern of reviewer 2 in detail, but are not sure that we have interpreted their concern correctly. Fusco and Minelli (2010) provide a comprehensive review of phenotypic plasticity in development and evolution. It is clear that there are no obvious 'alternative' phenotypes (i.e. polyphenic characters) in the expression of the frill or, indeed, any other cranial

region of *Protoceratops* in our dataset. Phenotypic plasticity may also be continuous (i.e. a monophenic character). This is reflected in the degree of trait expression, and hence variance, and thus expression could of course be determined by environmental factors, but this is exactly what is expected in socio-sexual traits that exhibit condition-dependence, as we have discussed. It is therefore not clear to us exactly how phenotypic plasticity of this kind is different from the condition-dependence that we expect to observe in a socio-sexual trait. We would of course be happy to revisit this point if we have misunderstood the reviewer's concern.

Refs.

Felice RN and Goswami A (2017). Developmental origins of mosaic evolution in the avian cranium. *PNAS*. doi: 10.1073/pnas.1716437115

Fusco G and Minelli A (2010). Phenotypic plasticity in development and evolution: facts and concepts. *Phil. Trans. R. Soc. B* (2010) 365, 547–556doi:10.1098/rstb.2009.0267